# FoxP2 isoforms delineate spatiotemporal transcriptional networks for vocal learning in the zebra finch

Zachary Daniel Burkett[1,2]*, Nancy F Day[1], Todd Haswell Kimball[1,3], Caitlin M Aamodt[1,4], Jonathan B Heston[1,4], Austin T Hilliard[5], Xinshu Xiao[1,2], Stephanie A White[1,2,4]

[1]Department of Integrative Biology and Physiology, University of California, Los Angeles, Los Angeles, United States; [2]Interdepartmental Program in Molecular, Cellular, and Integrative Physiology, University of California, Los Angeles, Los Angeles, United States; [3]Physiological Science Master's Degree Program, University of California, Los Angeles, Los Angeles, United States; [4]Interdepartmental Program in Neuroscience, University of California, Los Angeles, Los Angeles, United States; [5]Department of Biology, Stanford University, Stanford, Stanford, United States

**Abstract** Human speech is one of the few examples of vocal learning among mammals yet ~half of avian species exhibit this ability. Its neurogenetic basis is largely unknown beyond a shared requirement for FoxP2 in both humans and zebra finches. We manipulated FoxP2 isoforms in Area X, a song-specific region of the avian striatopallidum analogous to human anterior striatum, during a critical period for song development. We delineate, for the first time, unique contributions of each isoform to vocal learning. Weighted gene coexpression network analysis of RNA-seq data revealed gene modules correlated to singing, learning, or vocal variability. Coexpression related to singing was found in juvenile and adult Area X whereas coexpression correlated to learning was unique to juveniles. The confluence of learning and singing coexpression in juvenile Area X may underscore molecular processes that drive vocal learning in young zebra finches and, by analogy, humans.

DOI: https://doi.org/10.7554/eLife.30649.001

*For correspondence:
zburkett@ucla.edu

**Competing interests:** The authors declare that no competing interests exist.

## Introduction

The ability to learn new vocalizations is a key subcomponent of language. Complex behaviors such as human speech and birdsong are rarely monogenic in origin, making the attribution of their direct molecular underpinnings a challenge (*Marcus and Fisher, 2003*). While language is unique to humans, learned vocal behavior is present in a number of animal taxa. Among laboratory animals, the zebra finch songbird (*Taeniopygia guttata*) is the primary genetic model for vocal learning, and song learning in this species shares numerous parallels with human speech development. For example, both species share corticostriatal loops for producing vocalizations and have direct projections from cortical neurons onto brainstem motor neurons that control the vocal organs, a connection that is lacking or reduced in non-vocal learners (*Lemon, 2008*; *Jürgens, 2002*; *Arriaga et al., 2012*; *Doupe and Kuhl, 1999*; *Petkov et al., 2012*). The brains of avian vocal learners contain a distributed corticostriatal network of clustered cells devoted to vocal production learning, commonly referred to as the song control circuit, offering tractable targets for experimental manipulation. Despite their evolutionary distance, humans and zebra finches exhibit shared transcriptional profiles in key brain regions for vocal learning that are unique from surrounding brain areas and from the brains of non-vocal learning species (*Pfenning et al., 2014*).

**eLife digest** Songbirds, much like in humans, have a critical period in youth when they are best at learning vocal communication skills. In birds, this is when they learn a song they will use later in life as a courtship song. In humans, this is when language skills are most easily learned. After this critical period ends, it is much harder for people to learn languages, and for certain bird species to learn their song.

When birds sing every morning, the activity of a gene called FoxP2 drops, which causes a coordinated change in the activity of thousands of other genes. It is suspected that FoxP2 – and the changes it causes – could be a part of the molecular basis for vocal learning. FoxP2 is also known to play a role in speech in humans, and both birds and humans have a long and a short version of this gene. Previous research has shown that when the long version of the gene was altered so its activity would no longer decrease when birds were singing, the birds failed to learn their song. Moreover, humans with a mutation in the long version have problems with their speech. However, until now, it was not known if modifications to the short version had the same effect.

Burkett et al. investigated whether there was a noticeable pattern in the effects of FoxP2 before and after the critical period in a songbird. The analysis found that during the critical period, a set of genes changed together as young birds learned to sing. This particular pattern disappeared as the birds aged and the critical period ended. Burkett et al. confirmed that when birds had the long version of FoxP2 altered, they were less able to learn. However, changing the short version of FoxP2 had little effect on learning but led to changes in the birds' song.

The genetic pathways identified in the experiments are known to be present in many different species, including humans. Related pathways have also been found to play a role in non-vocal learning in organisms as distantly related as rats and snails. This suggests that they could be acting as a blueprint for learning new skills. Few treatments for language impairments have been developed so far due to poor understanding of the molecular basis for vocal communication. The findings of this study could help to create new treatments for speech problems in people, such as children with autism or people with mutated versions of FoxP2.

DOI: https://doi.org/10.7554/eLife.30649.002

The forkhead box P2 (FOXP2) transcription factor was the first gene shown to be important for vocal learning in both humans and songbirds. Forkhead box proteins are characterized by the presence of DNA-binding FOX domains (*Clark et al., 1993*) and FOXP subfamily members form homo- or heterodimers at zinc finger and leucine zipper domains in order to bind DNA. In humans, a heterozygous mutation in the FOX domain of FOXP2 causes a rare heritable speech and language disorder in a cohort known as the KE family (*Vargha-Khadem et al., 1998*; *Lai et al., 2001*), potentially by altering the subcellular localization of the molecule (*Vernes et al., 2006*). While the mutation disrupts vocal learning (*Marcus and Fisher, 2003*) and also vocalization in vocal non-learners (*Chabout et al., 2016*; *Castellucci et al., 2016*), multiple FOXP2 isoforms are endogenous to both songbirds and humans, including one that lacks the DNA binding domain (*Teramitsu and White, 2006*; *Bruce and Margolis, 2002*). This truncated variant is referred to as FOXP2.10+ because, although it lacks the FOX domain, it retains the dimerization domains plus an additional 10 amino acids that are not found in the full length form (FoxP2.FL).

Consistent with its lack of a FOX domain, in vitro assays of FOXP2.10+ indicate that it may regulate other FoxP2 isoforms (*Vernes et al., 2006*). Since it retains the dimerization domain, it has been hypothesized to act as a cytoplasmic sink, binding to other FOXP proteins and preventing their entry to the nucleus and interaction with DNA. Investigation of FoxP2 function in zebra finches has revealed remarkable parallels with humans. Similar *FoxP2* expression patterns occur in developing human and zebra finch brains (*Teramitsu et al., 2004*). In zebra finches, knockdown of *FoxP2* in the song dedicated striatopallidal nucleus, Area X, during vocal development impaired vocal mimicry of tutor songs (*Haesler et al., 2007*), much as the KE family mutation impairs speech. These observations indicate that functional FoxP2 is necessary for proper vocal learning, an inference supported by work in songbirds (*Haesler et al., 2007*; *Heston and White, 2015*).

The unique organization of song control circuit neurons enabled the discovery that FoxP2 is dynamically downregulated within Area X when zebra finches practice their songs, termed 'undirected' (UD) singing (*Teramitsu and White, 2006*; *Miller et al., 2008*; *Hall, 1962*; *Immelmann, 1962*; *Dunn and Zann, 1996*). This decrease in FoxP2 is accompanied by increased vocal variability (*Miller et al., 2010*; *Hilliard et al., 2012a*), thought to be a form of vocal exploration. Blockade of FoxP2 downregulation impaired birds' ability to induce variability in their songs. A poor learning phenotype emerged following FoxP2 overexpression (*Heston and White, 2015*) that was remarkably similar to that observed following FoxP2 knockdown (*Haesler et al., 2007*). Taken together, these results indicate that the dynamic regulation of at least FoxP2.FL, and thereby the behavior-linked up- and down-regulation of its transcriptional targets, is necessary for the proper learning of vocalizations. No specific role in vocal behavior has yet been attributed to the FoxP2.10 + isoform.

These observations pinpoint FoxP2 as a molecular entry point to the pathways underlying vocal learning. In adult birds, we previously used Weighted Gene Coexpression Network Analysis (WGCNA) to identify thousands of genes regulated by singing specifically in Area X (*Hilliard et al., 2012a*; *Langfelder and Horvath, 2008*). Since adult zebra finches sing stable, or crystallized, songs, the transcription patterns underlying vocal learning were not identified. Here we conduct a new study with two goals: (1) Determine whether FoxP2.10+ may play a role in vocalization and, (2) Manipulate FoxP2 isoforms in juveniles to generate a broad range of behavioral and transcriptional states upon which to apply WGCNA and thereby reveal learning-related gene modules. Toward the first goal, overexpression of FoxP2.10+ revealed a unique role for this truncated isoform in the acute modulation of vocal variability. Toward the second goal, overexpression of either GFP or one of the two FoxP2 isoforms created three distinct groups of juvenile birds: one that was good at learning and acutely modulating variability (GFP), one that was poor at learning and acutely modulating variability (FoxP2.FL), and one that was good at learning but injected stability into song (FoxP2.10+). We applied WGCNA to the Area X transcriptome of birds across this behavioral continuum and discovered striatopallidal coexpression patterns that were positively correlated to learning. These learning-related patterns were present in juvenile but not adult Area X. However, singing-driven coexpression patterns in Area X were largely preserved between juveniles and adults, suggesting that: (1) song production modules are independent of learning state and (2) the spatiotemporal co-occurrence of both song production and learning-related gene modules in juvenile Area X is fundamental to vocal learning.

## Results

### Virus-mediated overexpression of FoxP2 isoforms affects song learning and/or vocal variability

Adeno-associated viral (AAV) constructs were used to drive overexpression of FoxP2.FL or FoxP2.10 + in Area X of developing males (*Figure 1—figure supplement 1*). To verify isoform-specific overexpression, we used two riboprobes in *in situ* hybridization experiments: one antisense to a region common to both transcripts (mid probe) and one antisense to a region near the 3' end of FoxP2.FL (3' probe; [*Teramitsu and White, 2006*]; *Figure 1A*). Robust signals beyond endogenous/background levels were observed in the striatopallidum of both hemispheres using the mid probe but only in the hemisphere injected with the FoxP2.FL construct using the 3' probe (*Figure 1B*). These results indicate that each viral construct overexpressed its encoded *FoxP2* isoform and was thus suitable for bilateral injection into Area X of juvenile males at 35d. An additional cohort received AAV encoding GFP as a control. We quantified levels of *FoxP2* expression at 65d by performing qRT-PCR with a set of primers that amplifies a region common to both transcripts (*Haesler et al., 2007*; *Olias et al., 2014*) and another set specific to the FoxP2.10+ (see Materials and methods). The first primer set indicated that *FoxP2* levels were higher in birds injected with either construct relative to control levels. When quantified by the second primer set, we found elevated PCR product only in the animals injected with the FoxP2.10+ construct (*Figure 1C*). No overexpression was detected in the ventral striatopallidum (VSP; the zebra finch striatum is interspersed with pallidal-like cells and is separate from the pallidum [*Reiner et al., 2004*]) (*Figure 1—figure supplement 2*). Taken together,

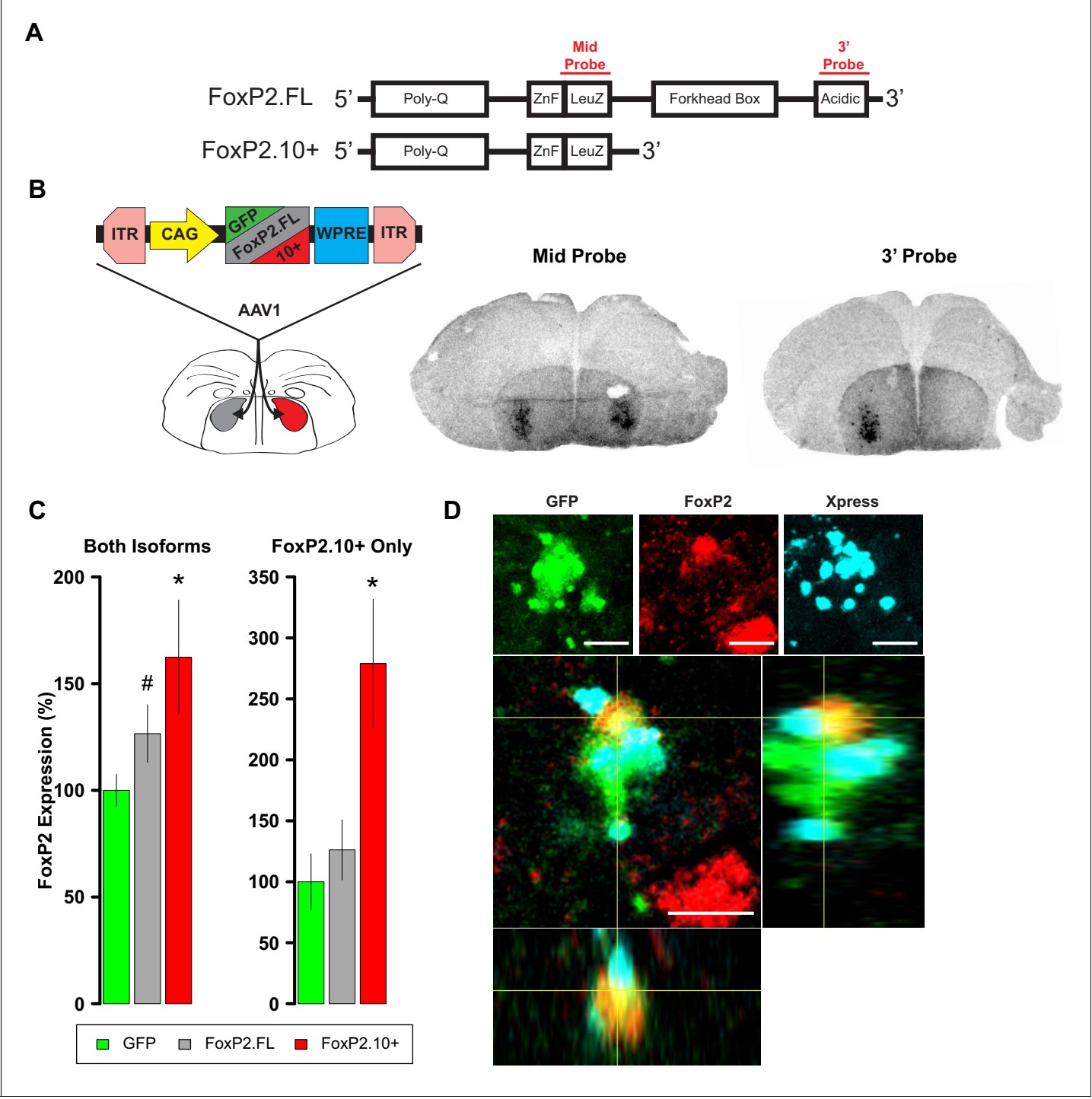

**Figure 1.** Overexpression of *FoxP2* isoforms. (**A**) Schematics show full-length (FoxP2.FL) and 10+ (FoxP2.10+) isoforms. Regions whose transcripts were targeted by the complementary riboprobes are shown in red. (**B**) Left panel depicts experimental design to test for isoform-specific expression in vivo. Middle and right images depict two sections from the same female brain. For purposes of validation only, the bird's right hemisphere (shown on left) was injected with an AAV expressing FoxP2.FL while the left hemisphere was injected with the FoxP2.10+ construct. Two weeks post-injection, robust signals were observed in the striatopallidum of both hemispheres using the mid probe but only in the hemisphere injected with the FoxP2.FL construct using the 3' probe. Signals reflect both the endogenous *FoxP2* expression pattern (**Teramitsu and White, 2006**; **Teramitsu et al., 2004**; **Teramitsu et al., 2010**) as well as enhanced levels due to viral-driven expression. (**C**) *FoxP2* expression quantified by qRT-PCR in juvenile males that were bilaterally injected with one of the constructs at 35d using primers that identify both isoforms (left graph) or only the *FoxP2.10+* isoform (right graph). Using the former primers, enhanced expression is observed in the FoxP2.FL (grey; 126.5 ± 13.53%; n = 6) and FoxP2.10+ (red; 162.4 ± 26.77%;

*Figure 1 continued*

n = 6) groups relative to levels of birds that received the GFP control construct (green; 100 ± 7.54%; n = 7). Using the 'FoxP2.10+ Only' primers, enhanced expression is only observed in the FoxP2.10+ group (red; 279 ± 52.69%; n = 6) vs. the FoxP2.FL (grey; 126.16 ± 24.61%; n = 6) and GFP (green; 100 ± 22.95%; n = 7). Values represent percentage relative to GFP ±SEM. * and # denote p=0.031 and p=0.084, respectively, of an unpaired two-tailed bootstrap test. (D) A cell in the zebra finch striatopallidum expressing GFP (indicating viral transduction; green), endogenous FoxP2 as revealed by an antibody directed to the C-terminus (red), and Xpress-FoxP2.10+ revealed by an antibody to the Xpress tag (cyan). The Xpress signal is reminiscent of FoxP2.10+ aggresomes observed by Vernes et al. (*Vernes et al., 2006*). Orthogonal views of the cell are presented below. Scale bar = 5 µM.

DOI: https://doi.org/10.7554/eLife.30649.003

The following source data and figure supplements are available for figure 1:

**Source data 1.** Contains averaged Ct values for each triplicate qPCR reaction presented in (C).
DOI: https://doi.org/10.7554/eLife.30649.006
**Source data 2.** Vector maps of AAV and HSV used in this study.
DOI: https://doi.org/10.7554/eLife.30649.007
**Figure supplement 1.** FoxP2 isoform protein expression.
DOI: https://doi.org/10.7554/eLife.30649.004
**Figure supplement 2.** FoxP2 qRT-PCR in VSP samples.
DOI: https://doi.org/10.7554/eLife.30649.005

these results indicate that both constructs were effective in elevating levels of their encoded FoxP2 isoform within Area X throughout the 30d experimental period.

Overexpression of a tagged form of FoxP2.10+ in a human neuronal cell line (SH-SY5Y) suggested that FoxP2.10+ acts as a posttranslational regulator of FoxP2.FL through heterodimerization and the formation of cytoplasmic aggresomes (*Vernes et al., 2006*). We thus examined the protein-level distribution of FoxP2.10+ and FoxP2.FL in the finch striatopallidum following overexpression of an N-terminus Xpress tagged FoxP2.10+ linked to a GFP reporter (see Stereotaxic Surgery and Viruses in Materials and methods). Transduced cells shared the distinctive FoxP2.10+ staining pattern of aggresomes seen previously. In FoxP2+ cells that co-expressed the Xpress tag and GFP reporter, endogenous FoxP2.FL signal was interspersed among Xpress-positive puncta (*Vernes et al., 2006*) (*Figure 1D*).

We previously found that, in unmanipulated birds, two hours of UD singing in the morning is sufficient to decrease Area X *FoxP2* mRNA (as measured by both the mid and 3' probes) and protein (*Teramitsu and White, 2006*; *Miller et al., 2008*). This decrease in FoxP2 was accompanied by an increase in the variability of UD songs, in the form of decreased self-similarity (see Materials and methods), that were sung subsequent to the two hour time-point, a paradigm which we term UD-UD (*Miller et al., 2010*; *Hilliard et al., 2012a*). In contrast, when birds were distracted from singing for two hours in the morning (non-singing; NS), their subsequent UD songs (termed NS-UD) were less variable. Moreover, overexpression of FoxP2.FL in Area X abolished the increase in vocal variability normally induced by the UD-UD paradigm (*Heston and White, 2015*). These observations indicate that downregulation of full length FoxP2 is important for acute vocal variability but we did not directly manipulate FoxP2.10+. Here, we performed similar behavioral experiments to test for the induction of vocal variability and included the FoxP2.10+ injected animals (*Figure 2A and B*). To assess whether UD singing drove an increase in vocal variability, we used the UD-UD paradigm (see Materials and methods) and quantified the effect of two hours of UD singing on the coefficient of variation (CV) of acoustic features in the subsequent UD songs of ~60d birds overexpressing GFP, FoxP2.FL, or FoxP2.10+. Results were compared to songs sung by the same birds undergoing the NS-UD paradigm. As predicted, GFP-expressing animals exhibited a negative effect size for most acoustic features, and FoxP2.FL overexpression diminished these practice-induced changes in vocal variability, replicating our previous findings (*Heston and White, 2015*) (*Figure 2C*).

Unexpectedly, in animals overexpressing FoxP2.10+, song variability after two hours of UD singing (UD-UD) was significantly *less than* that after two hours of non-singing (NS-UD) for syllable duration, amplitude modulation, and Wiener entropy (*Figure 2C*). Rather than increasing song variability (as in the GFP group) or creating a state of equivalent variability (as in the FoxP2.FL group), UD-UD singing led to markedly invariable songs in the FoxP2.10+ birds, suggesting a role for FoxP2.10+ in

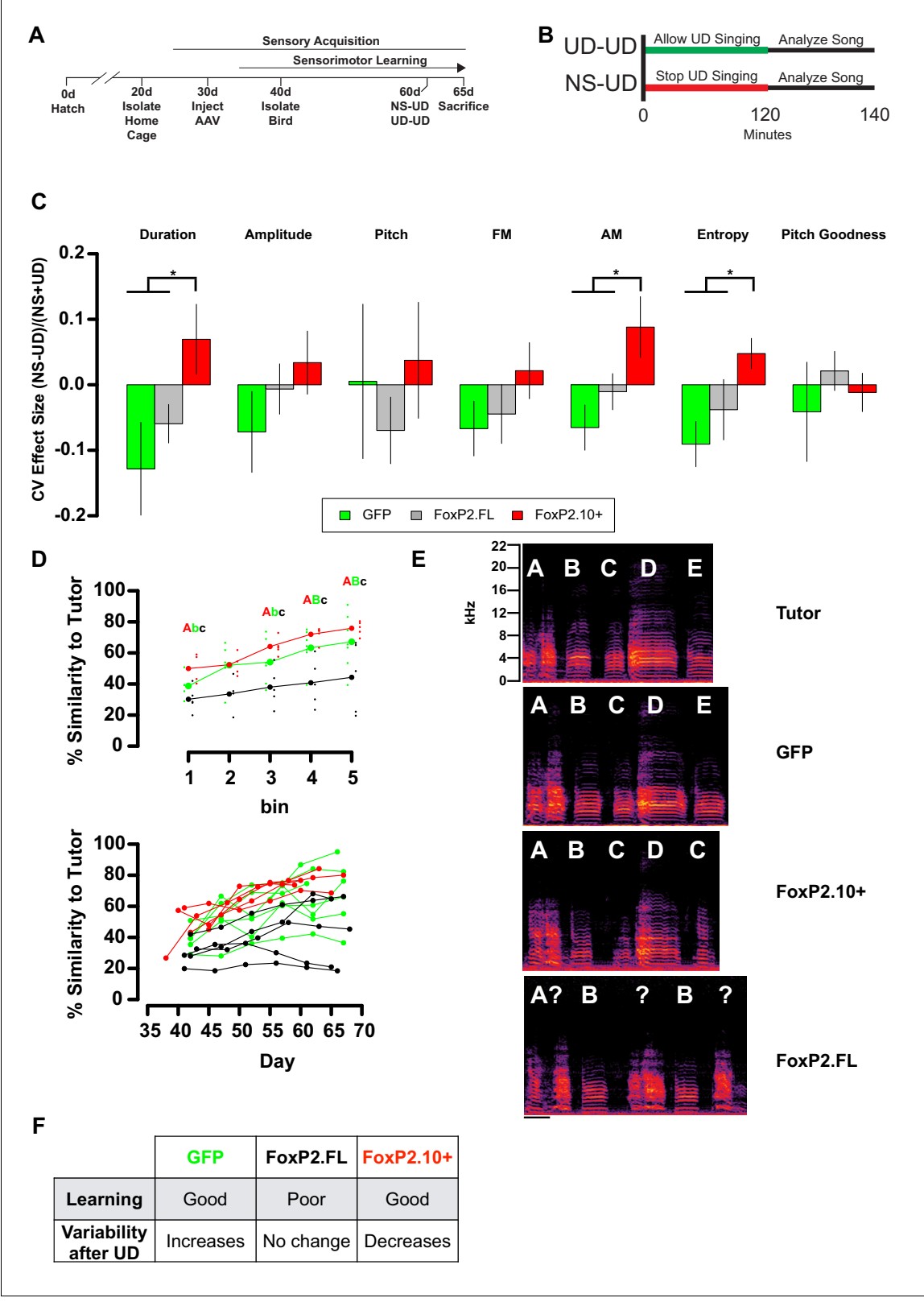

**Figure 2.** Overexpression of FoxP2 isoforms affect song learning and/or song variability. (**A**) Timeline of experimental procedures relative to critical periods in song development. (**B**) Schematic illustrates NS-UD or UD-UD experiments performed on adjacent days. (**C**) The effect size of two hours of UD singing on syllable CV was calculated using the formula (NS-UD)/(NS + UD) after an NS-UD, UD-UD experiment performed at ~60d and 61d as in (**B**). Overexpression of FoxP2.FL (grey bars; n = 16 syllables; Duration = −0.059 ± 0.029; AM = −0.010 ± 0.028; Entropy = −0.038 ± 0.04) diminishes

*Figure 2 continued on next page*

*Figure 2 continued*

singing induced variability relative to that seen in GFP-expressing controls (green bars; n = 9 syllables; Duration = −0.128 ± 0.071; AM = −0.065 ± 0.035; Entropy = −0.091 ± 0.034). In contrast, overexpression of FoxP2.10+ (red bars; n = 13 syllables; Duration = 0.070 ± 0.054; AM = 0.088 ± 0.047; Entropy = 0.048 ± 0.029) leads to a singing-induced state of relative invariability. Values and bar heights represent the average effect size for all syllables within the virus construct group ±SEM. * denotes significant result in one-way ANOVA (Duration: $F_{(2,35)}$ = 3.95, p=0.028; AM: $F_{(2,35)}$ = 3.96, p=0.028; Entropy: $F_{(2,35)}$ = 3.63, p=0.037) and Tukey's HSD post-hoc test (p<0.05). (D) Learning curves plot the relationship between percentage similarity to tutor as a function of time. Animals overexpressing GFP (green; letter 'B'; n = 7 birds;~65 d similarity = 67.2 ± 6.64%) or FoxP2.10+ (red, letter 'A'; n = 5 birds;~65 d similarity = 75.8 ± 2%) learn significantly better than those overexpressing FoxP2.FL (grey, letter 'C'; n = 5 birds;~65 d similarity = 44.3 ± 10.1%). Values are mean ±SEM. Data are binned by day (top panel; bold points represent group mean and shifted smaller points are individual birds) or by individuals (bottom panel). Significantly different groups tested by one-way ANOVA (Bin 1:~40d $F_{(2,11)}$ = 6.06, p=0.016; Bin 3:~55d $F_{(2,13)}$ = 6.04, p=0.014; Bin 4:~60d $F_{(2,14)}$ = 9.94, p=0.002; Bin 5:~65d $F_{(2,14)}$ = 4.76, p=0.026) and Tukey HSD post-hoc test (p<0.05) are denoted by capital and lowercase lettering. (E) Exemplar motifs of a tutor and three of his 65d pupils, each of which was injected with a different viral construct at 30d. These examples illustrate the percent similarity depicted in panel D. (F) Summary of the learning and variability phenotypes observed after virus injection.

DOI: https://doi.org/10.7554/eLife.30649.008

The following source data and figure supplement are available for figure 2:

**Source data 1.** Contains the effect sizes for each syllable that are presented in (C).

DOI: https://doi.org/10.7554/eLife.30649.010

**Source data 2.** Contains the binned similarity scores presented in the upper plot in (D).

DOI: https://doi.org/10.7554/eLife.30649.011

**Figure supplement 1.** Raw acoustic feature variability in the NS-UD and UD-UD conditions by virus group.

DOI: https://doi.org/10.7554/eLife.30649.009

promoting song stability. We also examined variability in the raw acoustic features of NS-UD and UD-UD song and found that expression of either FoxP2 isoform did not dramatically alter variability, indicating that the viral-driven overexpression specifically affected the modulation of variability (See 'Acute Modulation of Vocal Variability' in Materials and methods) and not its overall level (*Figure 2—figure supplement 1* and Materials and methods). Despite its suppressive effect on practice-induced song variability, overexpression of FoxP2.10+ did not impair overall vocal learning (*Figure 2D and E*). As shown by Heston and White (*Heston and White, 2015*), FoxP2.FL birds were capable of changing their songs over the course of the experiment (data not shown) but were less able to match their tutors' songs (*Figure 2D and E*). These results suggest that the ability to modulate between relatively low and high variability states is important for proper vocal learning.

In sum, our viral manipulations generated groups of animals in distinct states of vocal variability and learning. GFP-injected birds learned well and displayed singing-induced variability in the acoustic features of song. FoxP2.FL birds learned poorly and had no difference in their songs' acoustic variability following practice. FoxP2.10+ birds learned well but seemed to exist in a state where practice drives *in*variability in vocal acoustics. As such, a broad degree of both learning and variability induction existed across groups (*Figure 2F*). Next, we used these behavioral metrics as correlates to gene coexpression patterns to interrogate the transcriptional profiles underlying these traits.

## Gene modules in juvenile Area X that correlate to vocal behavior are enriched for communication and intellectual disability risk genes

We used RNA-seq to quantify gene transcription in Area X of 65d juveniles overexpressing GFP, FoxP2.FL or FoxP2.10+, then used WGCNA to identify gene coexpression modules and link them to song learning. We built an overall network composed from all samples together (*Figure 3A and B*), as well as construct-specific networks (*Figure 3—figure supplements 1–4*). In the overall network (see Materials and methods), 7461 genes formed 21 modules (*Figure 3A and B*, *Supplementary file 1*). We found significant correlations between module eigengenes and the following behaviors: tutor percentage similarity (i.e. vocal learning: darkred, green, and greenyellow modules), number of motifs sung (i.e. amount of singing: black, orange, darkgreen, royalblue, and blue modules), singing-induced acoustic variability (i.e. variability induction: black, brown, darkgreen, darkgrey, magenta, orange, pink, purple and turquoise modules), and motif identity (i.e. overall vocal variability: dark-grey module) (0.00008 < p < 0.05; *Figure 3B*). Hereafter, these modules are termed 'learning-related', 'song-production', 'variability-induction' and 'vocal variability' modules, respectively. We

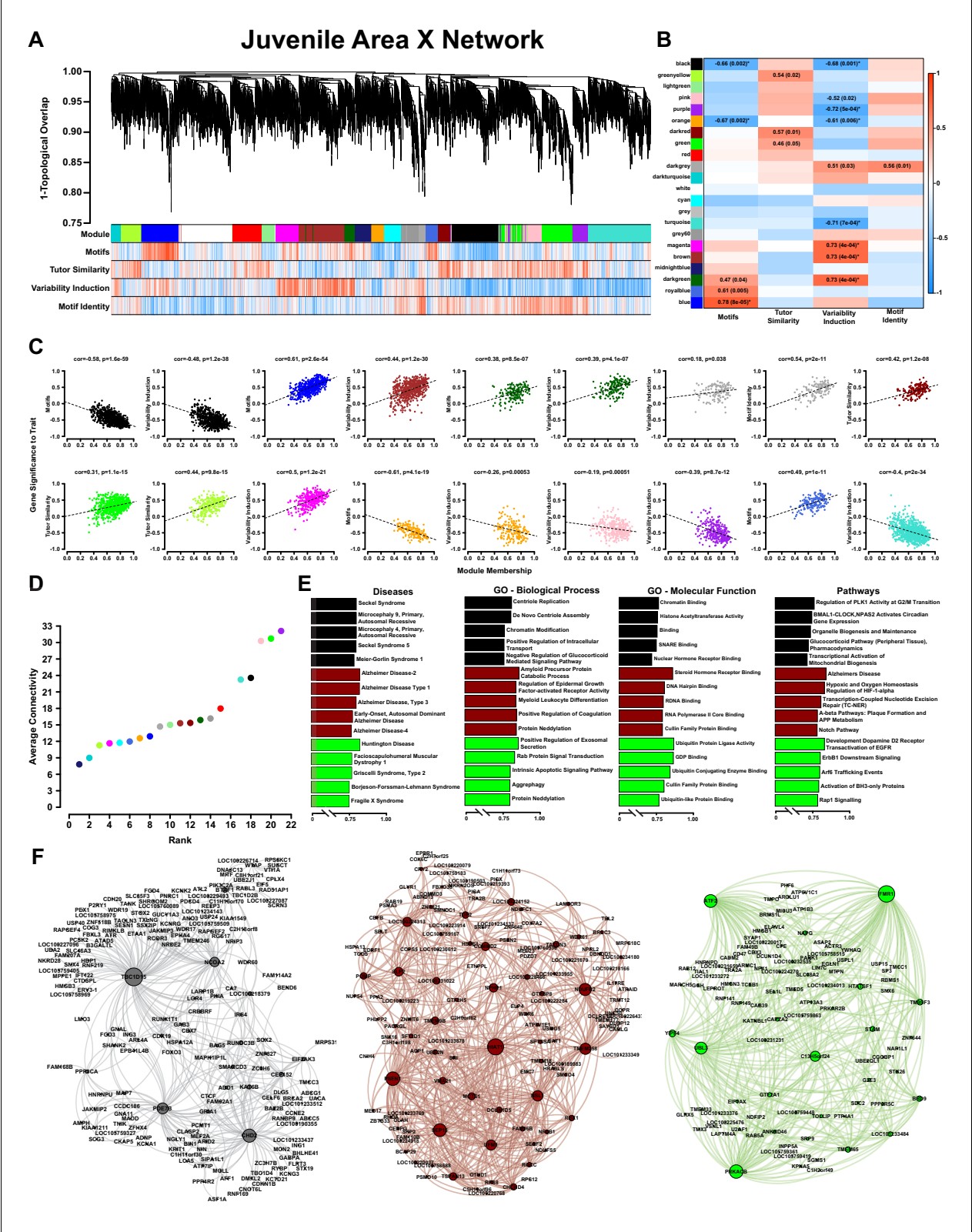

**Figure 3.** WGCNA yields behaviorally relevant modules. (**A**) Dendrogram (top) illustrates the topological overlap between genes in the juvenile Area X overall network. Modules delineated by automated tree trimming are shown below and are depicted by arbitrary colors. Beneath the color bar, gene significances to the quantified behaviors (number of motifs sung, tutor similarity, acute variability changes, and overall variability; see Results) are indicated by a heatmap wherein red indicates a positive correlation and blue indicates a negative correlation (see B for scale). (**B**) Correlations between

*Figure 3 continued on next page*

*Figure 3 continued*

module eigengenes and each behavior are presented as a heatmap. The Pearson's ρ and, in parentheses, Student's asymptotic p-values for modules where p≤0.05 are displayed. P-values are uncorrected for multiple hypothesis testing but those that pass FDR correction at p≤0.05 are denoted by * (See 'Correlation of behavior to gene expression' in Materials and methods). (C) For all significant module-trait correlations, the relationship between gene significance and module membership is plotted for each gene in the module. Dashed lines represent the linear regression and the Pearson's ρ ('cor') and p-value as determined by Fisher's z-transformation are indicated above each plot. (D) The average whole network connectivity (kTotal) within each module reveals that the purple, green, and pink modules are composed of the most strongly connected genes in the network. (E) Term significances for the black, darkred, and green modules are indicated for disease, gene ontology biological process and molecular function, as well as for pathways for categories annotated as 'neuronal' in the GeneCards GeneAnalytics software. (F) Network plots of the modules presented in panel E where nodes represent genes scaled by the node's intramodular connectivity and edge width displays the topological overlap between genes.

DOI: https://doi.org/10.7554/eLife.30649.012

The following source data and figure supplements are available for figure 3:

**Source data 1.** The behavioral metrics that were correlated to the network to generate (A) and (B).
DOI: https://doi.org/10.7554/eLife.30649.019

**Source data 2.** An R workspace containing the network presented in (A).
DOI: https://doi.org/10.7554/eLife.30649.020

**Source data 3.** An R workspace containing the processed expression data for the network presented in (A).
DOI: https://doi.org/10.7554/eLife.30649.021

**Source data 4.** The Pearson correlation values for each module eigengene and the behavioral metrics in *Figure 3—source data 1*.
DOI: https://doi.org/10.7554/eLife.30649.022

**Figure supplement 1.** GFP-only Area X network.
DOI: https://doi.org/10.7554/eLife.30649.013

**Figure supplement 2.** FoxP2.FL-only Area X network.
DOI: https://doi.org/10.7554/eLife.30649.014

**Figure supplement 3.** FoxP2.10+-only Area X network.
DOI: https://doi.org/10.7554/eLife.30649.015

**Figure supplement 4.** Module preservation between GFP vs FoxP2.
DOI: https://doi.org/10.7554/eLife.30649.016

**Figure supplement 5.** Juvenile Area X gene coexpression network.
DOI: https://doi.org/10.7554/eLife.30649.017

**Figure supplement 6.** Intersample correlation for Area X samples.
DOI: https://doi.org/10.7554/eLife.30649.018

examined all modules whose p-value was ≤0.05 and calculated the relationship between module membership and gene significance. (For definitions of WGCNA and network terms, see Materials and methods: WGCNA and network terminology. For information about significance levels reported here, see Materials and methods: Correlation of behavior to gene expression). For most modules, strong correlations were observed for each trait, indicating that the genes most representative of the module's overall expression profile were those most strongly related to the behavior (*Figure 3C*).

Connectivity is the core gene coexpression network concept and genes with high connectivity have the strongest coexpression relationships across the entire network, indicating greater importance to overall network structure and biological significance. The purple, green, and pink modules contained the most densely interconnected genes (*Figure 3—figure supplement 5*), and were correlated to percentage similarity to tutor (green learning-related module) or singing-induced variability (purple and pink variability-induction modules) (*Figure 3B and D*). These findings indicate that information about the relationships between gene coexpression and behavior was reflected in the structure of the network: A gene's relationship to a module or a module's relationship to the network was predictive of strong behavioral relevance. Therefore, we examined the most well-connected/hub genes within the context of their module (genes with the greatest intramodular connectivity) or the entire network (genes with the greatest whole-network connectivity). We discovered that many of these hub genes are known risk genes for human disease. For example, of the 7462 genes in the overall network, Fragile X Mental Retardation 1 (*FMR1*) had the third highest connectivity and was the most well connected member of the green module (*Supplementary file 1*). Deficiency in FMR1 gives rise to Fragile X Syndrome, a genetic disease with a multitude of symptoms including intellectual deficiency and speech and language impairment.

To attribute biological meaning to the modules, we calculated a module significance score for the resulting disease, gene ontology, and pathway annotations returned from GeneAnalytics (*Ben-Ari Fuchs et al., 2016*) (See Materials and methods). The top five terms for the black song production module (negatively correlated to the amount of singing), the brown variability induction module (positively correlated to variability induction), and green learning-related module (positively correlated to learning) are shown in *Figure 3E* with comprehensive results presented in *Supplementary file 2*. Since most modules contain hundreds of genes, prioritizing the ontology terms by the connectivity of their annotated genes allows genes with the greatest network importance (*Figure 3F*) to emphasize the terms with the greatest biological importance (*Figure 3E*).

## Juvenile Area X modules for learning, but not singing, are preserved in juvenile VSP

To validate the specificity of the Area X modules to vocal behavior, we compared the overall Area X network to a network constructed from the adjacent non-song VSP (*Hilliard et al., 2012a*; *Feenders et al., 2008*) from the same animals. Area X and VSP networks were constructed using the genes that were common to the two, enabling analysis using module preservation functions. We hypothesized that the genes in the Area X song production modules would have no correlation to behavior in VSP since, despite its close proximity and similar cell type composition, the VSP is not similarly linked into song control circuitry (*Person et al., 2008*). Moreover, a body of evidence suggests that the song control circuit evolved as a specialization of existing motor circuitry (*Pfenning et al., 2014*; *Feenders et al., 2008*; *Barrett, 2012*; *Oakley and Rivera, 2008*). As predicted, no module in the VSP network displayed any correlation to any of the singing or learning behaviors as gene significances using Area X and VSP expression data are markedly different (*Figure 4A*, X vs. V). We calculated module preservation statistics between the two brain regions and observed that the song production modules were among the most poorly preserved (*Langfelder et al., 2011*) across the two networks (*Figure 4B*, *Supplementary file 3*). This result indicates differential connectivity of song production module genes between Area X (*Figure 4C*, top) and VSP (*Figure 4C*, bottom), further underscoring that Area X is specialized for song. This lack of preservation was not the product of differential gene expression between the two regions (*Figure 4D*, top) but instead reflected altered connectivity among similar genes (*Figure 4D*, bottom). In striking contrast to the song production modules, the green learning-related module was strongly preserved in VSP (*Figure 4B*, *Figure 3B*), indicating a generalized learning-related coexpression state exists in the juvenile striatopallidum that is specialized for singing in Area X.

## Juvenile Area X modules for singing, but not learning, are preserved in adult Area X

To provide further context for the modules observed in our overall network and how they relate to learned vocalization, we compared them with prior data from adult zebra finch Area X (*Hilliard et al., 2012a*; *Hilliard et al., 2012b*). Our present network captures a point in zebra finch development when birds are actively learning how to improve their songs whereas in adulthood, the learning process has ended and adult songs are 'crystallized'. Contrasts between juvenile and adult networks highlight gene coexpression patterns that change between the two learning states, and inform their molecular underpinnings.

Our previous study in adults found multiple modules in Area X that were correlated to singing crystallized songs. We reasoned that if highly similar coexpression patterns were present in juveniles, then they would likely be unrelated to learning. In this case, the capacity to learn a song might be attributable to other genes and/or the relationships between them. To compare across studies, we built two new, age-specific networks composed of genes common to the two original networks, then computed gene significance scores for all genes in both networks. We found a remarkable correlation between gene significances to singing in juveniles and adults (*Figure 5A*), showing that genes in Area X shared similar relationships to singing, whether it be positive, negative, or nonexistent, independent of the animal's age and learning state. The replicated discovery of specific sets of song-production genes across studies and ages speaks to the profound effect that singing behavior has on gene transcription profiles within the song-dedicated basal ganglia.

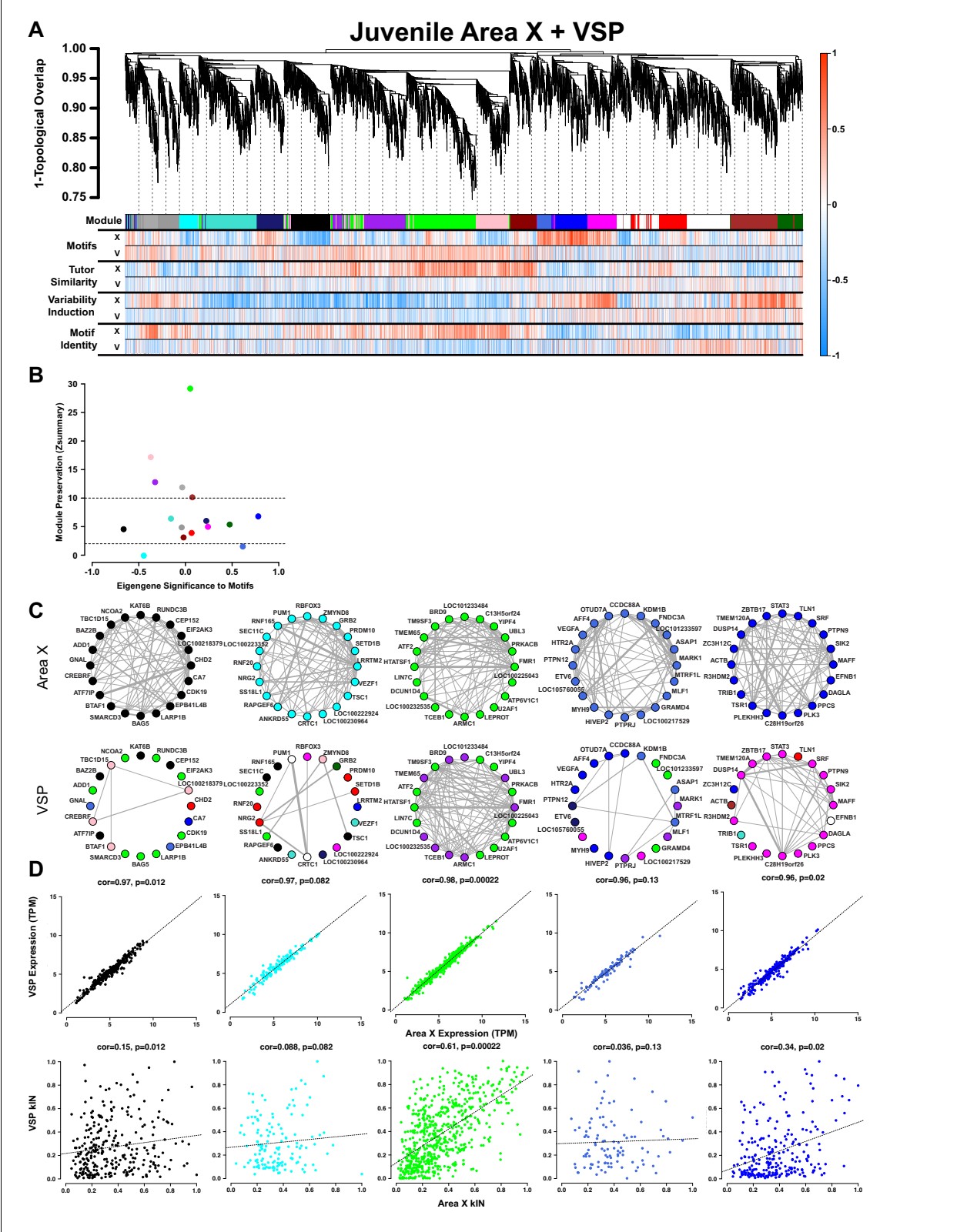

**Figure 4.** Juvenile Area X singing related gene coexpression patterns are not preserved in juvenile VSP. (**A**) Dendrogram (top) displays the topological overlap in Area X between genes common to both juvenile Area X and VSP networks. Beneath, the module assignments and the gene significances for each gene as calculated using expression from VSP ('V') or Area X ('X') for all behaviors are quantified as in *Figure 3A*. Module colors are consistent with those presented in *Figure 3*. (**B**) Module preservation (Zsummary) for all modules that were present in both Area X and VSP displayed as a function

*Figure 4 continued on next page*

*Figure 4 continued*

of module eigengene correlation to motifs. Lower and upper dashed horizontal lines indicate thresholds for low and high preservation, respectively. (**C**) Circle plots display the adjacencies between the 20 most well-connected genes in the Area X black, cyan, green, royalblue, and blue modules. The adjacency between genes is indicated by edge thickness. Genes grouped together in the black, cyan, royalblue, and blue song modules in Area X have numerous and strong connections. Those connections are weakened or nonexistent in VSP such that genes sort into different modules in VSP. In contrast, the green learning-related module genes maintain their common grouping and connections in VSP. (**D**) Raw gene expression is tightly correlated between Area X and VSP for the genes in the black, cyan, green, royalblue, and blue modules (top). Only the intramodular connectivity of the genes in the green learning-related module is correlated between Area X and VSP (bottom). Dashed lines represent the linear regression.
DOI: https://doi.org/10.7554/eLife.30649.023

The following source data is available for figure 4:

**Source data 1.** An R workspace containing the network presented in (**A**).
DOI: https://doi.org/10.7554/eLife.30649.024
**Source data 2.** An R workspace containing the processed expression data for the network presented in (**A**).
DOI: https://doi.org/10.7554/eLife.30649.025

We next calculated module preservation across the two studies, which assesses how well the coexpression relationships between genes persist across ages (*Langfelder et al., 2011*). We observed strong to very strong relationships between module preservation and correlation to singing, and genes related to singing clustered together independent of age (*Figure 5B and C*, *Supplementary file 4*). These results indicate that not only are the relationships between genes and singing consistent across ages but those genes' coexpression patterns are preserved as well.

Since singing-driven gene coexpression patterns were similar between juvenile and adult Area X, the capacity to learn vocalizations is not a product of large-scale differences in coexpression of the song production module genes. We therefore looked for any modules that differed between juvenile and adult Area X. We found that the green, greenyellow and darkred learning-related modules that were significantly correlated to tutor similarity in juveniles were poorly preserved in adult Area X (*Figure 5B and C*, *Supplementary file 4*). Irrespective of preservation between juvenile and adult Area X, the genes in song production and learning-related modules were similarly activated by singing (*Figure 5D*, top row) and the ranked gene expression within each module displayed a positive correlation across ages (*Figure 5D*, middle row). However, only the song production modules showed positive correlations between connectivity in juvenile and adult Area X (*Figure 5D*, bottom row). These results attribute the difference between juvenile and adult Area X not to differential expression or altered correlation to behavior, but to differential connectivity in adults of modules that are correlated to tutor similarity in juveniles. Our findings suggest that the capacity to alter vocalizations may not reside in the absolute expression level of a given gene but instead the gene's transcriptional context. For example, *FMR1* was poorly connected in the adult network but was positioned as a hub gene in the juvenile network, indicating the gene's importance during a developmental period when vocalizations are being actively modified but not during their maintenance. In general, genes that were positively correlated with learning and/or had high module membership in the green learning-related module had the greatest decrease in connectivity in adulthood (*Figure 5—figure supplement 1*).

## A bioinformatics approach indicates MAPK11 as an entry point to neuromolecular networks for vocal learning

Above we describe two classes of coexpression modules: (1) learning-related modules that are preserved throughout the striatopallidum but present only in juveniles, (2) song production modules that are preserved across age but specific to Area X. Therefore, song production modules and learning-related modules exist simultaneously only in juveniles, and their co-occurrence within Area X may reflect the capacity to dramatically alter vocalizations during sensorimotor learning. Therefore, we hypothesized that interactions between these two modules may drive the vocal learning process.

To test this idea using bioinformatics, we examined any genes linked to FoxP2, whose overexpression drove the broad range of tutor song copying in our animals. The gene with the greatest gene significance to learning was *MAPK11* (*Figure 6A and B*). Interestingly, in Foxp2 heterozygous knockout mice, *MAPK11* levels increase, supporting the interaction we observed here (*Enard et al., 2009*). To examine whether *MAPK11* could be a target of FoxP2 in the zebra finch, we scanned the

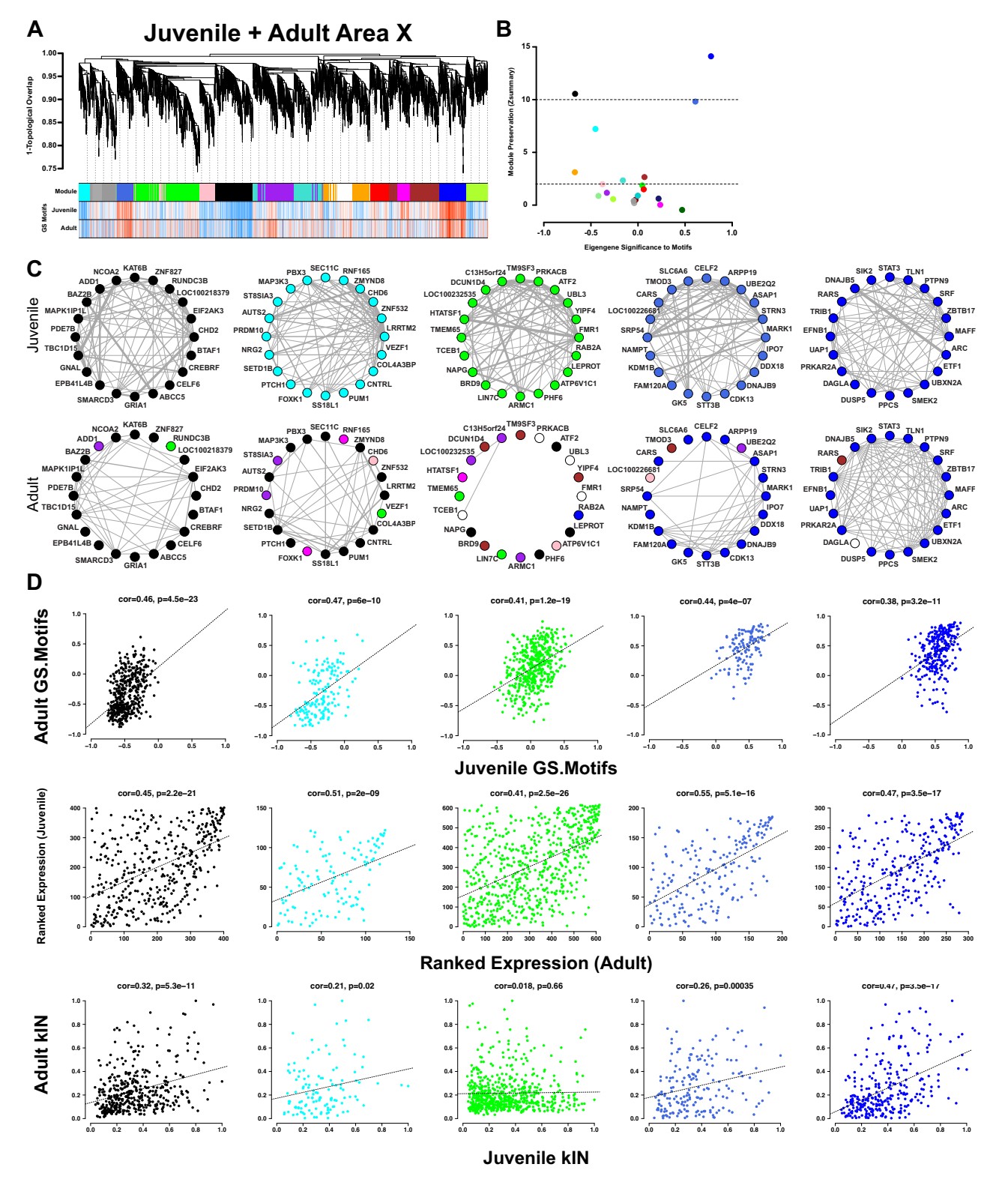

**Figure 5.** Area X song production but not learning-related modules are preserved into adulthood. (A) Dendrogram (top) displays the topological overlap in juvenile Area X between genes common to both juvenile and adult Area X networks. The module assignments and the gene significances to motifs in juveniles and adults are presented below. Module colors are consistent with those presented in *Figure 3*. (B) Module preservation (Zsummary) for all modules that were present in both juvenile and adult Area X displayed as a function of ME correlation to motifs. Lower and upper dashed

*Figure 5 continued on next page*

*Figure 5 continued*

horizontal lines indicate thresholds for low and high preservation, respectively. (**C**) Circle plots display the adjacencies between the 20 most well-connected genes in the juvenile Area X black, cyan, green, royalblue, and blue modules. The adjacency between genes are indicated by edge thickness. Genes grouped together in the black, cyan, royalblue, and blue song modules in Area X have numerous and strong connections that are mostly maintained in adulthood. The densely interconnected green learning-related module genes found in juveniles do not maintain these relationships in adulthood. (**D**) Strong positive correlations between gene significance to motifs exist for all modules (top row). Ranked expression values for the genes in each module also show positive correlation (middle row). Intramodular connectivity is more positively correlated between ages for the black, cyan, royalblue, and blue song production modules than for the green learning-related module (bottom row).

DOI: https://doi.org/10.7554/eLife.30649.026

The following source data and figure supplement are available for figure 5:

**Source data 1.** An R workspace containing the expression data and networks used to generate (**A**).
DOI: https://doi.org/10.7554/eLife.30649.028
**Figure supplement 1.** Differential connectivity as a function of green module membership.
DOI: https://doi.org/10.7554/eLife.30649.027

*MAPK11* gene for sequences corresponding to the FoxP2 binding motif from the JASPAR database (see Materials and methods) (*Nelson et al., 2013*; *Mathelier et al., 2016*). We found a match with a single base difference beginning 288 base pairs upstream of the zebra finch *MAPK11* transcription start site identified in the RefSeq model (*Figure 6C*). (Note that the RefSeq model may be incomplete; see *MAPK11* annotation note in Materials and methods). We then used chromatin immuno-precipitation followed by PCR (ChIP-PCR) to test whether or not FoxP2 binds this predicted *MAPK11* regulatory region. Chromatin-immunoprecipitation of FoxP2 enriched a *MAPK11* fragment of the predicted size and encompassing the putative FoxP2 binding site. Moreover, the sequenced fragment contains the FoxP2 binding motif (*Figure 6D*, *Figure 6—figure supplement 1*). Taken together, these data suggest that birds overexpressing FoxP2.FL may be limited in their capacity to learn due, at least in part, to FoxP2 regulation of *MAPK11*. In line with this, both the FoxP2.10+ and GFP animals had higher *MAPK11* gene significance scores for tutor similarity than did FoxP2.FL animals (*Figure 6A*).

A strength of WGCNA is the 'guilt by association' approach whereby genes in close network proximity to a gene of interest become candidates for a role in the same biological processes. With this in mind, we used *MAPK11* as an entry point to pathways related to vocal learning. We first scanned for genes with high topological overlap with *MAPK11* (e.g. the closest network neighbors to *MAPK11*). Many of these genes were well-connected members of the green learning module (*Figure 6E*). One such gene, *ATF2* (formerly known as *CREB2*), had the fifth highest green intramodular connectivity and third highest whole network connectivity (*Supplementary file 1*). ATF2 protein is necessary for proper development of the nervous system (*Reimold et al., 1996*) and serves a dual purpose in affecting transcription by binding to cAMP response elements and also by acetylating histones H2B and H4 (*Bruhat et al., 2007*; *Kawasaki et al., 2000*). Like *FMR1*, *ATF2* is poorly connected in the adult network (*Hilliard et al., 2012a*).

While its role in development of the nervous system has been defined, no specific relationship between ATF2 and learned vocalization has been described. In our network, the ATF2 acetylation target histone H2B sorted into the blue song production module, which is strongly and positively correlated to the act of singing (*Figure 3B*, *Supplementary file 1*) and acetylation of histone H2B at lysine five has been linked to learning and memory in rat hippocampus (*Bousiges et al., 2013*). A pathway such as this represents an interaction between a network hub in a learning module (*ATF2*) and a song production module gene (histone H2B) at a developmental time point at which the bird is actively learning its vocalizations.

To generalize this strategy, we used the Search Tool for the Retrieval of Interacting Genes/Proteins (STRING) database (*Szklarczyk et al., 2015*) to identify additional interactions between learning-related network hubs and song production genes in Area X. We submitted genes from the green, greenyellow, and darkred learning-related modules and the black, blue, darkgreen, orange, and royalblue song production modules, then filtered for cross-module interactions and scaled the confidence scores by the average intramodular connectivity of each gene in the interaction. This yielded a ranked list of interactions between genes positively correlated to learning and those correlated to singing, which was prioritized by weighted confidence score to yield the highest confidence

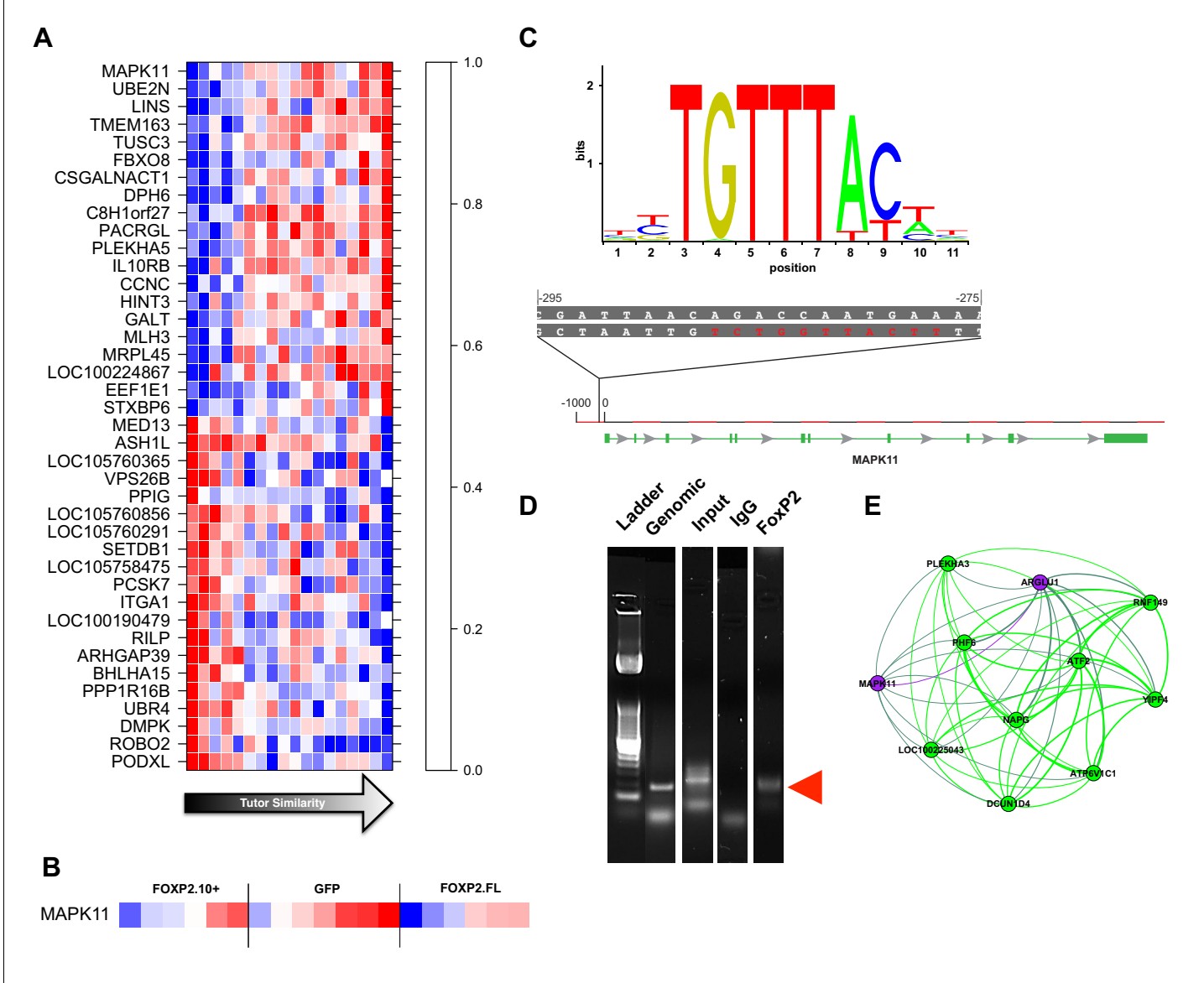

**Figure 6.** Gene significance and network position implicate MAPK11 as a molecular entry point to vocal learning mechanisms. (A) The 20 genes with the highest to lowest gene significances to tutor similarity (sorted from top to bottom) are shown. Each column represents a bird and columns are sorted in order of increasing tutor similarity from left to right. Gene expression is scaled such the highest and lowest expression across samples have the brightest shade of red or blue, respectively. (B) Expression of *MAPK11* is replotted, here separated by virus group and then sorted by increasing tutor percentage similarity. (C) The FoxP2 binding sequence as annotated by the JASPAR database (top) and a potential binding site found in the *MAPK11* 'promoter'. (D) Amplification of genomic DNA ('Genomic') with primers for a region of the MAPK11 'promoter' that contains a putative FoxP2 binding site enrich a fragment of predicted size (red arrowhead) in the pull-down lane (FoxP2) but not the control (IgG) lane. (E) MAPK11 and its 10 closest network neighbors, including green learning-related module members and hub gene ATF2, as defined by topological overlap.

DOI: https://doi.org/10.7554/eLife.30649.029

The following source data and figure supplement are available for figure 6:

**Source data 1.** The sorted gene expression data as presented in panel 6A.
DOI: https://doi.org/10.7554/eLife.30649.031

**Figure supplement 1.** *MAPK11* PCR Product Sequencing.
DOI: https://doi.org/10.7554/eLife.30649.030

interactions between genes with the greatest network importance (*Supplementary file 5*). These interactions were plotted as a network with proteins as nodes and interaction scores as edges (*Figure 7*). This approach allowed us to not only visualize the confidence in gene interactions but also the local neighborhoods formed by the protein interaction network, emphasizing genes of potentially greater importance in the vocal learning process based on the number of interactions they have.

We ranked interactions by four different metrics designed to emphasize or deemphasize gene significance, intramodular connectivity, and differential connectivity in juveniles vs. adults (see Materials and methods). These metrics provide a basis for selecting protein-protein interactions based on the relationship to the genes and their most strongly correlated behavior, the coexpression network importance of the genes, or the change in connectivity between juvenile and adult birds. In using the latter metric, the decreased connectivity of learning-related genes *ATF2* and *FMR1* in adulthood is accounted for and interactions involving those genes are prioritized. Interactions between *ATF2* and *IRF2*, *DUSP5*, and *FOS* are among the highest scoring interactions using this metric. All such interactions are presented in *Supplementary file 5*.

## Construct-specific networks

In addition to the overall Area X network presented above, we built and compared construct-specific networks from birds injected with the FoxP2.FL expressing virus versus those injected with the FoxP2.10+ expressing virus versus those expressing GFP (*Figure 3—figure supplements 1–3*). This analysis enabled us to assess the level of construct-driven changes in gene coexpression as well as to test for the presence of the learning-related module in the control birds whose FoxP2 levels were unmanipulated. We quantified module preservation between the FoxP2 networks and the GFP network (*Figure 3—figure supplement 4*). In both FoxP2 networks, a gradient of module preservation was observed versus the GFP network with both overlapping and significantly different modules observed. Birds in these experimental conditions were siblings, and in some cases from the same clutch, suggesting that the driving effect of network differences is the construct-specific manipulation. The green learning-related module was well-preserved across the three networks. The strong correlation of this module to learning passed false discovery rate correction in the GFP cohort comprised of only seven birds, indicating that the learning-related coexpression pattern observed in the overall network is also present without FoxP2 manipulation.

## Discussion

In this study, we overexpressed FoxP2 isoforms or GFP and thereby created a range of song learning and song variability induction (*Figure 2F*), ideal for transcriptome profiling and WGCNA. We constructed an overall Area X gene network and discovered modules correlated to singing, learning, and vocal variability. The network properties of these modules revealed strong relationships between gene module membership and the behavior(s) to which the modules were correlated.

To understand how gene coexpression patterns change across the boundary of the sensorimotor critical period for vocal learning, we compared the juvenile Area X overall network constructed here to one previously constructed from adult Area X (*Hilliard et al., 2012a*). We had competing hypotheses about whether the inability to learn new songs as an adult is resultant of changes to the song production modules observed in juveniles or associated with some other transcriptional change. Module preservation statistics revealed robust preservation of the juvenile Area X song production modules in the adult network, supporting the latter hypothesis. In striking contrast, the densely interconnected green learning-related module observed in juvenile striatopallidum was poorly preserved in adults, indicating that at least part of the learning-related transcriptome is altered by aging. Further, the green learning-related module was strongly preserved across the construct-specific networks (*Figure 3—figure supplements 1–4*) and robustly correlated to learning in the GFP network. This latter finding suggests that the coexpression of these genes occurs in non-manipulated birds and is not a byproduct of experimental perturbation of FoxP2 levels.

Because we created networks from VSP of the same animals, we could compare how well the Area X modules were preserved in a similar brain region that is unspecialized for song. As in *Hilliard et al. (2012a)*, Area X song production modules were poorly preserved in VSP in contrast to the strongly preserved green learning-related module. These experiments define juvenile Area X as

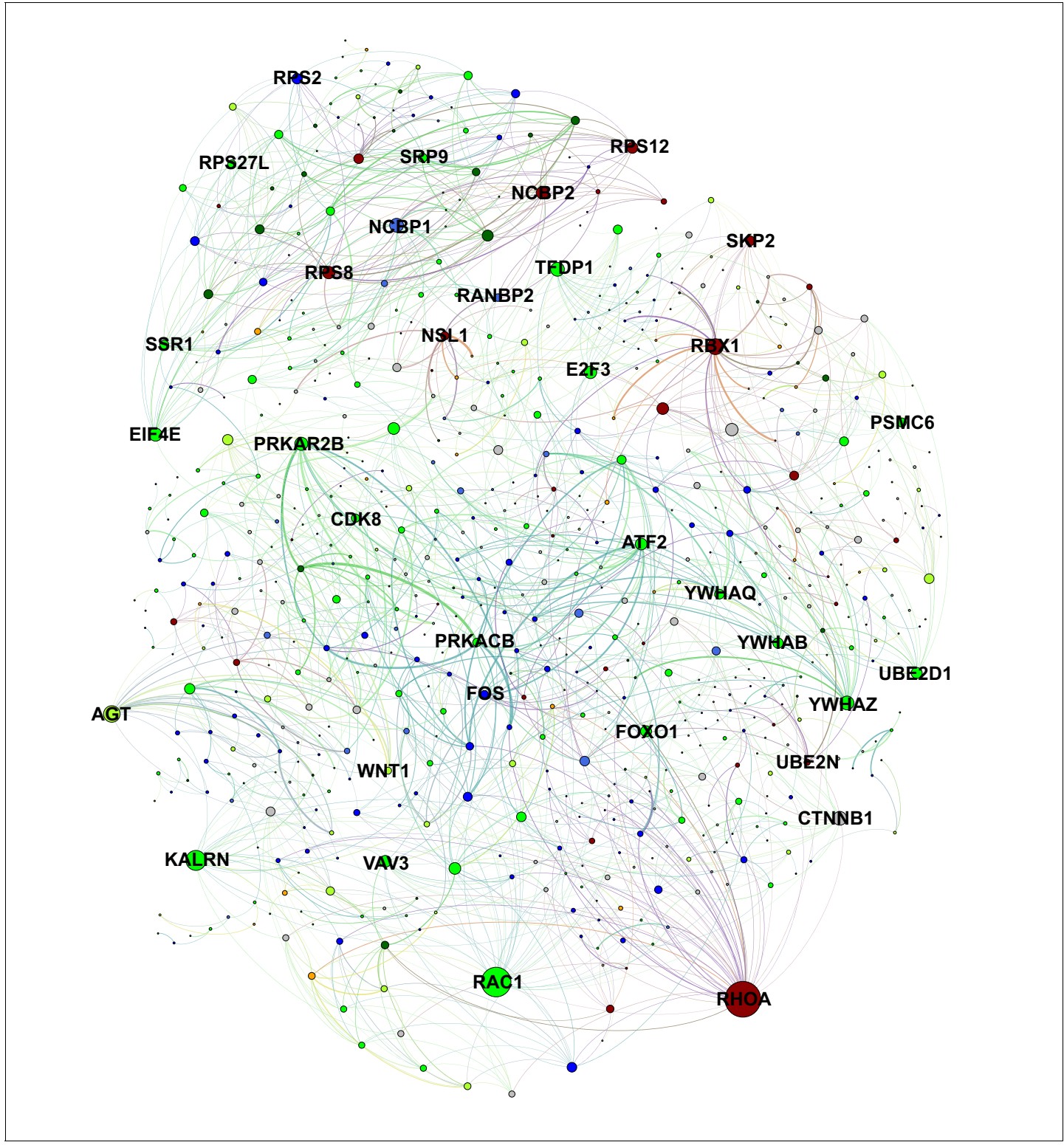

**Figure 7.** Protein-level interactions between song production and learning-related module genes in juvenile Area X. A protein interaction network plot using the STRING database between genes in learning-related (darkred, green, greenyellow) and song production (black, blue, darkgreen, orange, royalblue) modules. Nodes are scaled by number of connections. Edge width is determined by scaling the STRING protein interaction confidence score for the two nodes by the product of each node's intramodular connectivity. Interactions within learning or song production modules are omitted for clarity.

DOI: https://doi.org/10.7554/eLife.30649.032

*Figure 7 continued on next page*

*Figure 7 continued*

The following source data is available for figure 7:

**Source data 1.** An edgelist in. gexf format used to generate *Figure 7*.
DOI: https://doi.org/10.7554/eLife.30649.033

a nexus wherein the striatopallidal learning-related modules exist in tandem with song production modules. As the brain ages, singing continues to drive transcriptional patterns in Area X but the learning-related patterns are lost *(Figure 8A*; *Figure 8B*). Our findings suggest a model for the molecular basis of complex learned vocal behavior as – not specific genes or coexpression modules – but rather the spatiotemporal overlap of 'singing' and 'learning' building blocks. Song control nuclei are proposed to have evolved as specializations of pre-existing motor circuitry (*Pfenning et al., 2014*; *Feenders et al., 2008*). A similar principle may thus extend across the songbird telencephalon whereby nonspecialized/learning related and specialized/behavior related coexpression patterns converge to permit sensorimotor learning.

Our findings validate prior results in which overexpression of FoxP2.FL prevented practice-induced changes in song variability and impaired song learning. These results support the hypothesis that behavior-linked cycling of FoxP2, rather than its absolute level, is critical for vocal learning. In addition, we uncovered singing-induced vocal *in*variability as a novel behavioral effect of FoxP2.10 + overexpression. Despite the poor exploration of motor space induced by FoxP2.10+ overexpression, these animals learned their tutors' songs well, a finding seemingly at odds with motor learning theory where broad exploration of motor space is refined through practice before arriving at an 'ideal' precise pattern for execution of the skill (*Kaelbling et al., 1996*; *Wu et al., 2014*). A similar phenomenon was observed in a different species of passerine songbird, the Bengalese finch (*Lonchura striata domestica*), where two hours of UD singing resulted in *less* variable songs than those sung after two hour of non-singing (*Chen et al., 2013*). In both species, the inability to induce song variability did not affect vocal learning, suggesting that the ability to have relatively low or high variability states in singing are necessary to properly learn a song regardless of whether those differential variability states precede or follow singing.

WGCNA identified *FMR1* as a gene of great importance in a learning module. *FMR1* encodes an RNA-binding protein and therefore its levels could have a profound effect on a number of targets in the network (*Ascano et al., 2012*). FMR1 protein is expressed throughout the zebra finch song control circuit primarily in neurons, and birdsong has been suggested as an interesting model in which to study the gene's function (*Winograd and Ceman, 2012*; *Winograd et al., 2008*). Here, we observed a correlative link between *FMR1* expression and how well the animal copied its tutor's song, a novel association that could be reasonably hypothesized given the speech and language phenotype associated with FMR1 deficiency in humans. A key strength of WGCNA is the ability to query the network around genes known to be associated with a trait. *FMR1*'s close network neighbors included *ATF2* which has been associated with learning but has no prior link to vocal behavior. Further investigation into the learning-related modules is likely to reveal pathways fundamental to procedurally learned behavior.

To identify those molecules that may interact at this particular developmental time point and brain region, we selected *MAPK11* – a likely FoxP2 target (*Enard et al., 2009*) and the gene with the greatest significance to learning – to further investigate as an entry point to the pathways underlying learning behavior. Local neighborhood analysis of *MAPK11* in the coexpression network revealed high topological overlap with many strongly connected members of green learning-related module, including the hub gene *ATF2*. ATF2 is a phosphorylation target of MAPK11 and part of an evolutionarily-conserved pathway for learning and memory (*Guan et al., 2003*). This phosphorylation enhances ATF2 histone-acetyltransferase activity (*Enslen et al., 1998*; *Stein et al., 1997*). A known enzymatic substrate of ATF2 is histone H2B (*Kawasaki et al., 2000*), a member of the blue song production module that is positively correlated to singing. To probe for additional protein-protein interactions such as these, we mined the STRING database using song production and learning-related module members, then prioritized the interactions based on the network properties and/or behavioral significance of the input genes. A prioritized list of interactions and a complex network

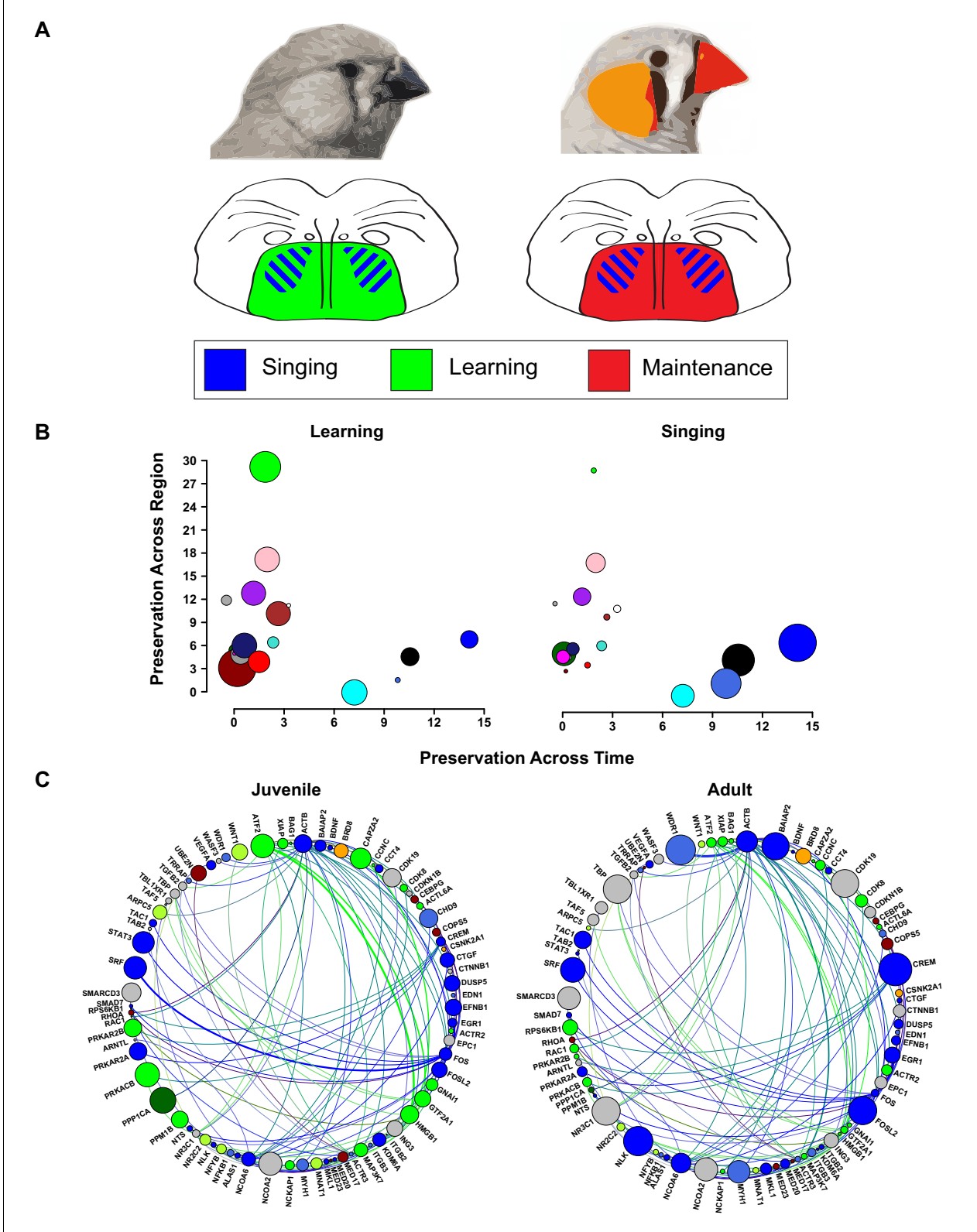

**Figure 8.** Changes in vocal plasticity state between juvenile and adult birds. (**A**) Schematics depict the juvenile straitopallidum (left) in a 'plastic' state in which genes in learning-related modules (green) are densely interconnected and of high importance in the network. Simultaneously, singing driven gene coexpression patterns (blue) occur in Area X. In the adult striatopallidum (right), song production modules (blue) exist as they do in juveniles, but the learning-related modules do not and are replaced by coexpression patterns that presumably underlie the maintenance of song (red). (**B**) Area X

*Figure 8 continued on next page*

*Figure 8 continued*

modules in the juvenile brain are plotted to emphasize their preservation in adult Area X (x-axis) and juvenile VSP (y-axis). Points representing the module colors are scaled by the module's absolute correlation to learning (left) or the absolute correlation to singing (right), emphasizing the preservation of singing coexpression patterns into adulthood and learning coexpression patterns in the juvenile striatopallidum. (C) Genes in song production or learning-related modules that are within two steps of ATF2 in the high-confidence protein interaction network are shown. Nodes are scaled by intramodular connectivity in juveniles (left) or adults (right) with edge width indicative of adjacency between genes in the coexpression network. The change in coexpression patterns across age groups causes decreased connectivity of many learning-related genes, driving an alteration in the network's landscape which may underlie the transition from song learning to song maintenance.

DOI: https://doi.org/10.7554/eLife.30649.034

emerged, highlighting genes based on their coexpression network importance and/or the number of protein level interactions in the database (*Figure 7*, *Supplementary file 5*).

While there are differences in overall gene expression between the juvenile and adult brain, the context within which genes express, that is, their connectivity, is drastically altered, especially in the learning-related modules. Changes in connectivity are not necessarily indicative of changes in the absolute level of a gene's expression, as evidenced by the comparisons between Area X and VSP (*Figure 4D*) or juvenile and adult Area X (*Figure 5D*), where expression levels correlate positively but connectivity does not. These data support the idea that the *coexpression* patterns, and thereby the genes' connectivity and network importance, contribute to the transition from a state of learning to a state of non-learning.

In using connectivity as a measure of network importance and protein interaction as a measure of functional biological output, the protein interaction landscape underlying learned vocal behavior shifts across the two developmental time points analyzed here. For example, the local interaction network around green module hub *ATF2* (defined as all those neighbors within two steps and with high confidence of protein interaction) is composed of well-connected genes in the learning-related and song production modules (*Figure 8C*, top). Moreover, the connections to learning-related genes are, themselves, inputs to well-connected network hubs. As the juvenile crosses over into adulthood, the connectivity of many of the learning-related genes, like *ATF2*, dramatically decreases. As part of the same process, the adjacencies between genes in the interaction network shift such that a connection to a learning-related gene is no longer one with a hub (*Figure 8C*, bottom). This shift in network importance may present a pattern underlying song maintenance rather than song learning, and potentially the closure of the critical period in which the bird can change its song.

To understand the mechanisms underlying the transition between the two learning states, our data highlight the importance of the network position of a gene. To enable vocal plasticity after critical period closure, a goal critically relevant to social and communication disorders, manipulations that coordinate gene expression such that poorly connected genes are reestablished as network hubs are likely required. Tools to accomplish a goal such as this do not yet exist, but the pathways prioritized and presented here provide a framework for teasing out testable components.

In sum, we have described the Area X transcriptome at a developmentally significant point in the vocal learning process and provided context for it in terms of aging and brain region specificity. We suggest numerous coexpression and protein level interactions that our data indicate are significant to vocal learning. Due to the large amount of data generated by this study, we provide interactive graphics describing the coexpression and protein interaction networks as a supplement to the figures and tables in the manuscript. These, and the compiled descriptive statistics are hosted at (https://www.ibp.ucla.edu/research/white/genenetwork.html). We encourage exploration of these datasets to confirm or refute their validity and to provide the molecule-to-behavior links suggested herein.

## Materials and methods

**Key resources table**

| Reagent type (species) or resource | Designation | Source or reference | Identifiers | Additional informa |
|---|---|---|---|---|

*Continued on next page*

*Continued*

| Reagent type (species) or resource | Designation | Source or reference | Identifiers | Additional informa |
|---|---|---|---|---|
| Genetic reagent (*Taeniopygia guttata*) | AAV1-FoxP2.FL | Virovek (Hayward, CA, USA), DOI: 10.1523/JNEUROSCI.3715-14.2015 | | |
| Genetic reagent (*Taeniopygia guttata*) | AAV1-FoxP2.10+ | Virovek (Hayward, CA, USA), this paper | | |
| Genetic reagent (*Taeniopygia guttata*) | AAV1-GFP | Virovek (Hayward, CA, USA), DOI: 10.1523/JNEUROSCI.3715-14.2015 | | |
| Genetic reagent (*Taeniopygia guttata*) | HSV-FoxP2.10+ | McGovern Institute for Brain Research at the Massachusetts Institute of Technology, this paper | | |
| Antibody | FoxP2 (goat polyclonal) | Abcam (Cambridge, MA, USA) | Abcam Cat# ab1307; RRID: AB_1268914 | ChIP: 4 ug |
| Antibody | FoxP2 (rabbit polyclonal) | ThermoFisher (Rockford, IL, USA) | Thermo Fisher Scientific Cat# 720031; RRID: AB_2610345 | ChIP: 4 ug |
| Antibody | FoxP2 (mouse monoclonal) | Santa Cruz Biotechnology (Dallas, TX, USA) | Santa Cruz Biotechnology, Cat# sc-517261; RRID: AB_2721204 | ChIP: 4 ug |
| Antibody | IgG (rabbit polyclonal) | EMD Millipore (Burlington, MA, USA) | Millipore Cat# 12-370; RRID: AB_145841 | ChIP: 4 ug |
| Antibody | Xpress (mouse monoclonal) | ThermoFisher (Rockford, IL, USA) | ThermoFisher Scientific Cat# R910-25; RRID: AB_2556552 | |
| Sequence-based reagent | FoxP2.FL F | Sigma Aldrich | Oligonucleotide CCTGGCTGTGAAAGCGTTTG | |
| Sequence-based reagent | FoxP2.FL R | Sigma Aldrich | Oligonucleotide ATTTGCACCCGACACTGAGC | |
| Sequence-based reagent | FoxP2.10+ F | Sigma Aldrich | Oligonucleotide CGCGAACGTCTTCAAGCAAT | |
| Sequence-based reagent | FoxP2.10+ R | Sigma Aldrich | Oligonucleotide AAAGCAATATGCACTTACAGGTT | |
| Sequence-based reagent | GAPDH F | Sigma Aldrich | Oligonucleotide AACCAGCCAAGTACGATGACAT | |
| Sequence-based reagent | GAPDH R | Sigma Aldrich | Oligonucleotide CCATCAGCAGCAGCCTTCA | |
| Sequence-based reagent | MapK11 F | Sigma Aldrich | Oligonucleotide CCCTTTCCCCAAATGGCAGA | |
| Sequence-based reagent | MapK11 R | Sigma Aldrich | Oligonucleotide TATGAGCCTTGCCTTGGAGC | |
| Sequence-based reagent | Mid probe | DOI: 10.1523/jneurosci.1662-06.2006 | | |
| Sequence-based reagent | 3' probe | DOI: 10.1523/jneurosci.1662-06.2006 | | |
| Commercial assay or kit | ChIP-IT High Sensitivity | Active Motif (Carlsbad, CA, USA) | Active Motif Cat# 53040 | |
| Commercial assay or kit | Qiagen RNeasy Micro | Qiagen (Germantown, MD, USA) | Qiagen Cat# 74004 | |
| Commercial assay or kit | Illumina TruSeq Stranded Poly-A Prep | Illumina (San Diego, CA, USA) | Illumina Cat# 20020594 | |
| Software, algorithm | VoICE | DOI: 10.1038/srep10237 | RRID: SCR_016004 | |

*Continued*

| Reagent type (species) or resource | Designation | Source or reference | Identifiers | Additional informa |
|---|---|---|---|---|
| Software, algorithm | STAR | DOI: 10.1093/bioinformatics/bts635 | RRID: SCR_015899 | |
| Software, algorithm | SAP | DOI: 10.1006/anbe.1999.1416 | RRID: SCR_016003 | |
| Software, algorithm | WGCNA R Package | DOI: 10.1186/1471-2105-9-559 | RRID: SCR_003302 | |

## Subjects

All animal use was in accordance with NIH guidelines for experiments involving vertebrate animals and approved by the University of California, Los Angeles Chancellor's Institutional Animal Care and Use Committee. Birds were selected from breeding pairs in our colony.

## Experimental timeline

The experimental timeline is schematized in *Figure 2A*. Breeding cages that contained candidate experimental birds were placed in sound attenuation chambers along with their parents and siblings when juveniles reached ~20 d, as in Heston and White (*Heston and White, 2015*). Chambers were continuously recorded so as to capture tutor song. At 30d, juvenile males were bilaterally injected with AAV1 into Area X to overexpress either FoxP2.FL, FoxP2.10+, or GFP, then returned to their chambers. At 40d, juvenile males were isolated from all other birds and continuously audio-recorded. At ~60 d, an 'NS-UD' experiment was performed according to the methods of Miller et al., Chen et al., and Heston et al. (*Heston and White, 2015*; *Miller et al., 2010*; *Chen et al., 2013*) to assess the induction of vocal variability. On the 'NS-UD' day, for the first two hours after lights-on, birds were distracted by gentle 'shushing' if they attempted to sing. (Those that sang >10 motifs were excluded from that day's experiment). On the 'UD-UD' day, birds were allowed to sing UD song for the first two hours after lights-on. The level of variability in songs sung subsequent to those two hours was quantified.

At 65d, birds were sacrificed following two hours of UD singing with one exception: In order to assure a broad range of song amounts immediately preceding sacrifice (and thereby capture a range of singing-induced gene expression), we distracted one bird in the GFP group from singing during the two hours preceding sacrifice.

A total of 19 birds received stereotaxic injections with AAV (7 GFP, 6 FoxP2.FL, 6 FoxP2.10+). Sample size was based on numbers used in Heston and White (*Heston and White, 2015*) where 5–8 animals per group were sufficient to reveal treatment effects. The authors of the WGCNA R package recommend a minimum of 15 samples for building a network (https://labs.genetics.ucla.edu/horvath/CoexpressionNetwork/Rpackages/WGCNA/faq.html), so we ensured at least five animals in each of the three groups.

## Song recording

Countryman EMW or Shure SM93 omnidirectional lavalier microphones were used to continuously record birds from ~20 d until sacrifice (65d). Sounds were digitized using PreSonus FirePod or PreSonus Audioboxes at a 44.1 kHz sampling rate and 24-bit depth. Recordings were managed by SAP 2011 software (*Tchernichovski et al., 2000*).

## Stereotaxic surgery and viruses

### Behavior and RNA-seq experiments

As described in *Heston and White (2015)*, 30d juvenile males were anesthetized using 2–4% isoflurane in pure oxygen and secured in a custom-built avian stereotaxic apparatus, then injected with virus bilaterally into Area X at the following coordinates: 45° head angle, 5.15 mm rostral of the bifurcation of the midsagittal sinus, 1.60 mm lateral of the midline, and to a depth of 3.3 mm. Virus was injected via a Drummond Nanoject II through a glass microelectrode (~40 μM inner diameter) backfilled with mineral oil. Three 27.6 nL injections were performed with a 15 s wait between injections and a 10 min wait before retraction of the electrode so as to minimize vacuum action pulling the virus away from the injection site. Incisions in the scalp were closed with Vetbond (3M, St. Paul, MN, USA). Birds received oxygen for ~2 min until alert, then returned to their home cages.

AAV1 used in Heston and White (*Heston and White, 2015*) and produced by Virovek (Hayward, CA) was used here. AAV1s contained zebra finch FoxP2.FL or FoxP2.10+ coding sequences (*Teramitsu and White, 2006*) (Genbank Accession Number DQ285023), or that for GFP, downstream of the CMV early enhancer/chicken β actin (CAG) promoter. Virus titers were all ~2.24E + 13 vg/ml, thus equivalent volumes were delivered to each bird irrespective of construct. Heston et al. (*Heston and White, 2015*) estimated that $24 \pm 5.5\%$ of neurons at the epicenter of the virus injection are transduced and that $96.7 \pm 1.7\%$ of cells that are transduced are neurons. These transduction rates are sufficient to observe a behavioral effect of the virus and were thus used in the present study.

## Histological assessment of FoxP2.10+ overexpression

FoxP2.10+ is a naturally occurring truncated isoform of FoxP2.FL, with a unique 10 amino acid sequence at its C-terminus. There is currently no antibody specific to this truncated isoform, presenting a challenge to its immunological detection. The limited cloning capacity of AAV precluded our ability to express a reporter gene in the viruses that we used for behavioral and RNA-seq experiments. Moreover, we opted not to include an epitope tag on AAV-expressed FoxP2 isoforms in order to avoid any conformational changes that could confound our behavioral or RNA-seq analyses. For histological analysis only, however, we took advantage of the larger cloning capacity of HSV to express FoxP2.10+ tagged with an Xpress epitope at its N-terminus downstream of the IE 4/5 promoter and a GFP transduction reporter downstream of the CMV promoter (McGovern Institute for Brain Research at the Massachusetts Institute of Technology, Cambridge, MA). Surgical procedures were identical to those performed with AAV except that the virus was diluted to 60% in PBS immediately preceding injection, per the manufacturer's recommendation. HSV reaches peak expression more rapidly than does AAV, thus HSV-injected birds were sacrificed 3–5 days post-injection (*Neve et al., 2005*).

## *In situ* hybridization

*In situ* hybridizations were performed as in *Jacobs et al. (1999)* using two [$^{33}$P]UTP-labeled riboprobes antisense to distinct regions of zebra finch *FoxP2* (*Teramitsu et al., 2004*). 20 µM thick sections were thaw-mounted onto Superfrost Plus microscope slides (ThermoFisher Scientific, Waltham, MA, USA), then postfixed with 4% paraformaldehyde in PBS, pH 7.4.

## PCR primers

To quantify levels of *FoxP2.FL*, we selected a primer pair previously used to quantify FoxP2 knockdown (*Haesler et al., 2007*; *Olias et al., 2014*). The forward sequence was 5'-CCTGGCTG TGAAAGCGTTTG-3' and the reverse was 5'ATTTGCACCCGACACTGAGC-3'. We designed a primer pair for *FoxP2.10+* using the NCBI Primer-BLAST tool (*Ye et al., 2012*). The input sequence was *FoxP2.10+* mRNA CDS (GenBank accession DQ285023.1). The forward primer sequence was 5'-CGCGAACGTCTTCAAGCAAT-3' and the reverse sequence was 5'-AAAGCAATATGCACTTACAGG TT-3'. Primer specificity was determined by obtaining a single peak in melting curve analysis and obtaining a single amplicon of predicted size following qPCR. *GAPDH* forward and reverse primers were 5'-AACCAGCCAAGTACGATGACAT-3' and 5'-CCATCAGCAGCAGCCTTCA-3', respectively.

## qRT-PCR experiments

200 ng of RNA from Area X micropunches was reverse transcribed into cDNA using the Bio-Rad iScript cDNA Synthesis Kit (Hercules, CA, USA). 25 µL qPCR reactions were assembled in MicroAmp Optical 96-Well Reaction Plates (ThermoFisher Scientific). Reaction components were 0.5 µL cDNA, 200 nM primers, 12.5 µL PowerUp SYBR Green Master Mix (ThermoFisher Scientific), and 10.75 uL nuclease-free water. Cycling conditions were 50℃ for 2 min, 95℃ for 2 min, then 40 cycles of 95℃ for 15 s and 60℃ for 1 min. A dissociation step of 95℃ for 15 s, 60℃ for 1 min, 95℃ for 15 s, and 60℃ for 15 s was then performed. All reactions were run in triplicate and all samples for an individual animal were run together on the sample plate. *FoxP2* expression was quantified relative to *GAPDH* and normalized to the GFP-injected animals using the $2^{-\Delta\Delta CT}$ method (*Livak and Schmittgen, 2001*).

## Immunostaining

For histological analyses, animals were sacrificed 3–5 days following HSV injection then perfused with warm saline followed by ice cold 4% paraformaldehyde in 0.1 M phosphate buffer. Tissue was cryosectioned at 20 μM, thaw-mounted onto glass microscope slides, and stored at −80°C until use. Thawed sections were incubated overnight with goat-anti-FoxP2 (1:500; Abcam, Cambridge, UK; [*Thompson et al., 2013*]) and mouse-anti-Xpress (1:500; ThermoFisher Scientific, Waltham, MA). AlexaFluor 546 donkey-anti-goat (1:500) and AlexaFluor 405 donkey-anti-mouse (1:250) secondary antibodies were used to generate anti-FoxP2 and anti-Xpress signals, respectively. Sections were visualized using a Zeiss (Oberkochen, Germany) LSM 800 confocal microscope and processed using NIH ImageJ (*Schneider et al., 2012*).

## Song analysis and statistics

### Motif similarity

The Similarity Batch in SAP was used to quantify the acoustic similarity between pupil and tutor songs (*Tchernichovski et al., 2000*). Asymmetric comparisons were performed between 10 tutor motifs (obtained from the final day before the pupil was acoustically isolated) and 20 pupil motifs (obtained every ~3 days following viral injection). We used the average percentage similarity from these comparisons as a representative of how well the pupil learned its tutor's song on a given day of analysis. Statistical significance of motif similarity data was calculated by performing one-way ANOVAs on the average percentage similarity score of each animal across virus groups within each time bin, as depicted in *Figure 2D*. If the ANOVA yielded a significant result, Tukey's Honest Significant Difference (HSD) was used as a post-hoc test.

### Overall vocal variability

To broadly assess the amount of variability in the animal's song preceding sacrifice, asymmetric comparisons between 20 pupil motifs and themselves were conducted. We calculated the motif identity for all motif-motif comparisons as the product of their percentage similarity and accuracy divided by 100. Higher identity scores indicate lower variability within the batch.

### Acute vocal variability modulation

For finer-grained analyses of acoustic variability as presented in *Figures 2C* and *Figure 2—figure supplement 1*, we utilized SAP and Vocal Inventory Clustering Engine (VoICE; [*Burkett et al., 2015*]; https://github.com/zburkett/VoICE). Syllables from the first 20 min following two hours of non-singing or undirected singing on the NS-UD experiment days were hand segmented, had their acoustic features quantified in the SAP Feature Batch, then clustered by VoICE. Data for analyses of acoustic features were taken from the VoICE output. Effect sizes were calculated using the formula (NS-UD)/(NS +UD), where values were the CV of a given acoustic feature following two hours of NS or UD. Thus, negative values indicate increased song variability after UD singing (see below for more information regarding this transformation). Statistical significance for each song feature was assessed by one-way ANOVA on the CV effect size for all syllables from all animals within each group. Tukey's HSD was used as a post-hoc test in the instance of a significant ANOVA result. For the raw acoustic data, as presented in *Figure 2—figure supplement 1*, the syllables were considered paired within virus construct and across singing context. Paired T-tests were used to assess whether two hours of non-singing vs. two hours of undirected singing significantly altered the CV for each acoustic feature.

## Song analysis: (NS-UD)/(NS + UD) effect size vs. raw acoustic feature CV

The calculation of effect size was performed because it allows for comparison across virus groups instead of a series of paired comparisons within group (*Miller et al., 2015*). The transformation normalizes acoustic features so that any observed changes are viewed in the context of the initial values. We present a hypothetical example in the table below where a change of 50 Hz for two syllables is given a greater weight for a syllable that has an overall lower frequency when using the transformation we applied for our song data:

| Syllable A | | | | Syllable B | | | |
| --- | --- | --- | --- | --- | --- | --- | --- |
| NS | UD | Raw delta | (NS-UD)/(NS + UD) | NS | UD | Raw delta | (NS-UD)/(NS + UD) |
| 100 Hz | 150 Hz | 50 Hz | −0.2 | 500 Hz | 550 Hz | 50 Hz | −0.048 |

## Tissue collection and processing, RNA extraction, cDNA library preparation, and sequencing

Two hours following lights-on at ~65 d, birds were sacrificed by decapitation. Brains were rapidly extracted and frozen on liquid nitrogen, then stored at −80°C until all brains were collected. As in *Hilliard et al. (2012a)*, tissue micropunches of Area X and VSP were performed. Brains were coronally sectioned on a cryostat at 30 μM until Area X became visible. Area X and outlying VSP were punched using a 20 gauge Luer adapter and stored in RNAlater (Qiagen, Germantown, MD) at −80°C until RNA extraction was performed. 30 μM sections were then collected, thaw mounted, and thionin stained for post-hoc validation of punch accuracy.

Total RNA extraction was performed as in *Hilliard et al. (2012a)*. Samples were processed semi-randomly and in parallel with another sequencing project. Tissue punches from both studies were processed in batches of 8. We used Qiagen RNeasy Micro Kits (Cat. No 74004) following the manufacturer's protocol and QIAzol as the lysis reagent. An additional wash beyond the manufacturer's protocol was performed in RW1 and RPE buffers. Final elution volume was 20 μL. Extracted total RNA were stored at −80°C until all RNA extractions were completed. All extractions were completed over the course of two weeks.

Total RNA was provided to the UCLA Neuroscience Genomics Core (UNGC; https://www.semel.ucla.edu/ungc) where RNA quality was assessed on an Agilent TapeStation (Agilent Technologies, Santa Clara, California). RNA of sufficient quality (RIN >8) was then used to generate cDNA libraries using the Illumina TruSeq Stranded Poly-A Prep Kit (Illumina, San Diego, CA, USA Cat No 20020594). Libraries for each sample were divided across two lanes and sequenced in a total of 8 lanes using an Illumina HiSeq 2500 in high output mode, generating between 15 and 35 million 50 bp paired-end reads per library.

## RNA-seq preprocessing and WGCNA

Raw FASTQ files furnished by UNGC were first quality controlled using FASTQC (http://www.bioinformatics.babraham.ac.uk/projects/fastqc/). FASTQC returned results indicating high quality across all bases in each read in each sample and no adapter contamination was detected, therefore we did not perform any filtration of the reads before alignment. Reads were aligned to the NCBI zebra finch genome assembly 3.2.4 (http://www.ncbi.nlm.nih.gov/assembly/524908/) and RefSeq annotations using STAR (*Dobin et al., 2013*). Mismatch tolerance was two base pairs. Only uniquely mapped reads were considered in downstream analyses. The featureCounts() function in the Rsubread R package was used to count all reads mapping within exon features, then all exon counts were summed to the gene level so that each gene had a single value of reads mapped to it (*Liao et al., 2014*; *Liao et al., 2013*). Gene expression was then quantified by calculation of transcripts per million (TPM). TPM values were log2 transformed and genes with zero variance across samples were removed. We checked for batch effect on average expression resultant of RNA extraction group, RNA extraction experimenter, and across sequencing lanes. No batch effects were observed. We used an iterative process of removing gene expression data from single samples whose expression was >2.5 SD of that gene's expression across all samples, repeating until no samples remained with expression >2.5 SD away from the gene's average expression across all samples. Finally, we calculated the intrasample correlation (ISC) and used a hard cutoff of 2 SD away from the group ISC for removal of samples from the study. No sample in any group (Area X or VSP) was >2 SD from the group ISC. Data were quantile normalized as the last step. Final data input to WGCNA was 13665 and 13781 genes for Area X and VSP networks, respectively, across 19 total samples.

We calculated the soft thresholding power for construction of the WGCNA adjacency matrix using the pickSoftThreshold function in the WGCNA R package at 18 for Area X and 14 for VSP. We then constructed a signed network using the blockwiseModules function in the WGCNA R package. For the Area X network, we used a minimum module size of 100 genes and deepSplit was set equal

to four for Area X and two for VSP. Genes were required to have at least a connectivity of 0.3 with their module eigengene in order to remain a member of their module and the module 'core' (=minimum module size/3) needed to have a minimum eigengene connectivity of 0.5 for the module to not be disbanded. All other parameters were set to default. Networks were iteratively constructed with genes in the grey module removed from the expression data after each round of network building and module definition. The networks were considered final after no genes were placed into the grey module.

During network construction, FoxP2 was removed, presumably due to the lack of coexpression with other genes in the network resulting from virus-driven overexpression. Therefore, we added FoxP2's expression data back into the final overall network and it became the only gene in the grey module. Once coexpression modules were defined, we correlated vocal behavior to the module eigengenes. Since the grey module included only a single gene with no significant behavioral correlations, it was excluded from module-trait analyses.

## WGCNA and network terminology

WGCNA is a well-established technique for gleaning biologically relevant clusters of coexpressed and functionally related genes from microarray and sequencing data. WGCNA methods and terminology are summarized and defined in numerous manuscripts (*Hilliard et al., 2012a*; *Zhang and Horvath, 2005*; *Dong and Horvath, 2007*; *Zhao et al., 2010*; *Yip and Horvath, 2007*; *Horvath, 2011*). For the sake of convenience, we provide working definitions of network terms that we use throughout the manuscript. Definitions of greater detail are available in the manuscripts cited above.

- Adjacency (a): The first step of network construction is to generate an adjacency matrix where $A_{ij} = S_{ij}^{\beta}$, where i and j are genes, S is the expression correlation across samples, and $\beta$ is an empirically derived power to which the correlation is raised such that the resulting network approximates a scale free topology.
- Connectivity (k): Connectivity is a measure of connectedness of a given gene, either in the context of its module (kIN) or the entire network (kTotal). Connectivity is defined as follows: $k_i = \sum_{j=1}^{N} a_{ij}$ where i and j are genes, N is all of the genes in the module or network, and a is the adjacency between genes i and j.

- Topological overlap: Adjacency is transformed to topological overlap as a method of calculating the interconnectedness (or similarity) between two nodes. Topological overlap is defined as follows: $\omega_{ij} = \frac{l_{ij}+a_{ij}}{\min\{k_i,k_j\}+1-a_{ij}}$ and $l_{ij} = \sum_{u\neq i,j} a_{iu} a_{uj}$, where u represents all genes besides i and j. A and k are defined above.
- Gene significance: The Pearson correlation between a gene's expression profile and, in our work, a given behavioral metric.
- Module eigengene: The first principal component of a module's gene expression profile, a method of summarizing an entire module in one vector.
- Module membership: The correlation between an individual gene expression profile and a module eigengene. Genes with high module membership tend to have high intramodular connectivity and are referred to as intramodular hubs. Of note, genes can have high module membership in more than one module.
- Zsummary: Along with median rank, a term for quantifying preservation of gene coexpression patterns between two independent datasets (*Langfelder et al., 2011*), such as between juvenile and adult Area X or juvenile Area X and juvenile VSP. Zsummary is a composite preservation score defined as the average of Zdensity and Zconnectivity, which assess the preservation of connection strength among network nodes (e.g. Are strongly connected nodes in one network also strongly connected in the other?) and the connectivity patterns between nodes (e.g. Do the patterns of connection between specific nodes exist in both networks?), respectively, following permutation tests under the null hypothesis. Higher Zsummary scores indicate better preservation.

## Correlation of behavior to gene expression

Calculation of gene significance to a trait requires the definition of a single value to which the amount of gene expression in each sample is correlated. Gene significances were calculated for the

following traits: Motifs, defined as the number of motifs each animal sang in the two hours following lights-on on the day of sacrifice; Tutor similarity, defined as the percentage similarity between the pupil and its tutor on the day of sacrifice; Variability induction, defined by inserting Wiener entropy CV scores into the equation (NS-UD)/(NS + UD) from the first twenty syllable renditions sung during the NS-UD experiment performed at ~60 d; Motif identity, defined as the product of the similarity and accuracy scores divided by 100 of the last 20 motifs sung by each bird before sacrifice. Song variability was assessed on the motif level for the purpose of gene significance calculations so as to obtain a single value for each animal.

Following network construction, modules were summarized by calculating a module eigengene, defined as the first principal component of the module's expression data using the moduleEigengenes() function in the WGCNA R package. The relationship between a module and a behavior was assessed by determining the Pearson correlation between the module eigengene and continuous behavioral traits as defined in 'Song Analysis and Statistics', above. Significance was then determined by calculating the Fisher transformation of each correlation using the corPvalueFisher() function in the WGCNA R package. We performed p-value corrections for module-trait correlations using the p.adjust() function with the number of comparisons equal to the number of traits (4) by the number of modules (21; the FoxP2-only grey module was not included for purposes of p-value correction). The p-values presented in this manuscript are uncorrected for multiple hypothesis testing but those that pass FDR-correction at p<0.05 are indicated. We chose to present uncorrected p-values due to the small sample size used to create the overall network (n = 19 birds). The authors of WGCNA suggest a minimum of 15 samples with >20 preferred (https://labs.genetics.ucla.edu/horvath/CoexpressionNetwork/Rpackages/WGCNA/faq.html). P-value corrections drive nearly all results to insignificance, including well preserved module-trait relationships that are present in adults and survive such corrections due to the larger sample size in that study. We use the significant but uncorrected p-values in this study as a guide toward interesting module-trait relationships, then use the properties of the network to inform the downstream analysis.

Our choice of behavioral traits for correlation to the gene network was hypothesis-driven. In addition to the obvious quantification of vocal learning, the comparison for variability induction was planned, as indicated by the fact that we conducted the NS-UD and UD-UD behavioral paradigms (prior to the bird's sacrifice) that led to it. We originally used these paradigms as a method for naturally regulating FoxP2 levels, before we had identified a virus that was effective in doing so. In that study (*Miller et al., 2010*), our prediction was that behavioral conditions that lead to low endogenous FoxP2 in Area X (namely 2 hr of UD singing), would be associated with higher levels of variability. This was indeed the case. We replicated this finding in zebra finches (*Heston and White, 2015*) but did not observe the same phenomenon in Bengalese finches (*Chen et al., 2013*) as noted in our Discussion. The feature highlighted by those studies was Weiner entropy.

## Gene ontology, module significance, and term significance

At the time of this study, annotation of the zebra finch genome is relatively sparse, thus zebra finch gene symbols were converted to their Human Genome Organisation (HUGO) Gene Nomenclature Committee (HGNC) paralogs, then submitted to GeneAnalytics, a comprehensive tool for the contextualization of gene set data that integrates across multiple databases (*Ben-Ari Fuchs et al., 2016*). Genes with no known human homolog were excluded. Symbols were submitted to the GeneCards GeneAnalytics suite at http://geneanalytics.genecards.org (*Ben-Ari Fuchs et al., 2016*). GeneCards enrichment scores were converted into p-values, which were used as the input to module significance calculations. Module significance of a term was defined as the product of the average module membership for each gene annotated with a term, and one minus the p-value for that term such that the genes with the highest module membership and lowest p-value prioritize the terms (*Hilliard et al., 2012a*). Term significance was defined by weighting the module significance score by the gene significance for a given behavioral metric.

## Transcription factor binding site analysis

The FoxP2 consensus binding sequence from the JASPAR database (*Nelson et al., 2013*; *Mathelier et al., 2016*) was converted into a position-weight matrix (PWM) and used to scan the promoter (defined as the first 1000 base pairs upstream of the transcription start site in the RefSeq

models) for each gene in the zebra finch genome. Putative FoxP2 binding sites were identified using the matchPWM function in the Biostrings R package (https://bioconductor.org/packages/release/bioc/html/Biostrings.html) with a minimum hit score of 80%.

## MAPK11 annotation note

The MAPK11 region discussed in this manuscript was identified using methods described above. Upon closer inspection of the MAPK11 RefSeq annotation model, we believe the identified region does not lie within the promoter but instead within an intronic region of MAPK11. There is currently no experimental evidence to verify the RefSeq model's predicted transcription start site and the Ensembl model for MAPK11 is considerably longer (323 residues vs. 285 residues) due to an expanded N-terminus region. Further, the chicken MAPK11 RefSeq model is 361 residues and contains an N-terminus residue (MSERGGFYRQELNKTVWEVPQRYQNLTPVGSGAYGSVC) that maps ~12 kb upstream of the second exon of chicken MAPK11. This residue does not map to the zebra finch genome, presumably because a gap in the genome exists ~13 kb upstream of zebra finch MAPK11. The MAPK11 N-terminus peptides of other songbird species (Bengalese finch, starling, white-throated sparrow, great tit and Tibetan ground-tit) are highly similar to that in chicken and align to the first exon in chicken MAPK11. This peptide is found in mice and humans, indicating high conservation. We thus posit that the MAPK11 RefSeq annotation in zebra finch is incomplete on the 5' end and that we are reporting a binding site internal to MAPK11 and not at the promoter.

## Chromatin immunoprecipitation-PCR

Chromatin immunoprecipitation (ChIP) was performed using ChIP-IT High Sensitivity (Active Motif, Carlsbad, CA, USA, Cat. No. 53040) following the manufacturer's protocol. Whole brain was isolated from an adult male zebra finch, minced, and crosslinked in a formaldehyde solution. The tissue was homogenized with a hand-held tissue homogenizer for 45 s at 35,000 rpm. Following homogenization, the sample was sonicated at 25% amplitude 30 s on, 30 s off, for 10 min. A portion of the sonicate was de-crosslinked and quantified. The sample was split evenly into three tubes. A cocktail of anti-FoxP2 primary antibodies were applied to one sample (Millipore, Billerica, MA, USA Cat. No. ABE73, ThermoFisher Scientific Cat. No. 5C11A2, and Abcam ab16046), IgG in another (Millipore 12–370), and the third was input DNA. After an overnight incubation, the samples were washed, de-crosslinked and subjected to PCR.

The 'promoter' sequence for *MAPK11* was binned into 100 bp regions for primer construction. *MAPK11* primers were as follows: forward 5'- CCCTTTCCCCAAATGGCAGA-3' and reverse 5'-TATGAGCCTTGCCTTGGAGC-3'. PCR protocol was performed using DreamTaq PCR Master Mix per manufacturer's protocol. A PCR protocol was used as follows: (1) 95°C 1 min, (2) 95°C 30 s, (3) 67°C 30 s, (4) 72°C 1 min, repeat (2-4) for 40 cycles, (5) 72°C 10 min. PCR products were run on a 1.5% agarose gel in the presence of SYBR Safe to allow visualization of DNA. PCR products were purified (QIAQuick Gel Extraction Kit) and sent for sequencing by Laragen, Inc. Reverse primers sent for sequencing are as follows: 5'-TATGAGCCTTGCCTTGGAGC-3' and 5'-CCTATGAGCCTTGCCTTGGA-3'.

## Protein interaction networks and scaling of interaction confidence scores

STRING is a comprehensive database of known and predicted protein-protein interactions derived from experimental data, coexpression data, automated text mining, and also pulls information from other interaction databases. STRING accepts gene symbols as input, then mines for interactions between those genes and assigns a confidence score between 0 and 1 based on the evidence in the database for the genes' interaction. We submitted gene symbols for the human homologs of module members to STRING then operated on the highest confidence interactions ($\geq$0.9) in downstream analyses.

Interaction scores were scaled by different metrics to emphasize or deemphasize network position and/or relationship to behavior (*Supplementary file 5*). Those metrics are:

1. The product of each gene's connectivity in juvenile Area X network: emphasizes interactions between the most connected genes in the juvenile network.

2. The product of each gene's differential connectivity between juvenile and adult Area X networks: emphasizes interactions between genes that are of high network importance in juveniles but not adults.
3. The product of each gene's gene significance for learning or singing: emphasizes interactions between genes that are strongly correlated to behavior independent of their connectivity.
4. The product of each gene's connectivity and gene significance: emphasizes interactions between genes that are strongly correlated to behavior and of highly connected in the juvenile network.

## Network visualization and interactive figures

Network plots presented in this manuscript were constructed using the freely available plotting software, Gephi (https://gephi.org), using edge lists prepared in R and exported in the. GEXF format.

We have created interactive versions of many of the network plots in this manuscript (*Figure 3F*) all additional Area X modules (similar to *Figure 3F* but not presented in the manuscript), and the protein interaction network presented in *Figure 7*. They are hosted at our laboratory website (https://www.ibp.ucla.edu/research/white/genenetwork.html) along with high resolution static PDF versions. Interactive figures were exported from Gephi using the Sigma.js Exporter plugin (https://github.com/oxfordinternetinstitute/gephi-plugins).

In weighted coexpression networks, each node (i.e. gene) is connected to every other node in the network, even if the weight of the edge (i.e. connection) is zero. Therefore, plots depicting nodes and their edges with other genes become exceedingly complicated and unintuitive if all nodes and edges are included. In an effort to sparsify the networks and present the most salient data, we removed edges and genes from the coexpression networks using the following workflow: first, remove ≤98% of edges, then remove all disconnected nodes, then remove all nodes that are not part of the network's main component (e.g. the largest group of connected nodes). The remaining nodes and edges were plotted.

In this manuscript, we present three types of network plots that look similar but convey different data. The three types are as follows:

1. The overall gene coexpression network, as in *Figure 3—figure supplement 5* and https://sites.google.com/a/g.ucla.edu/genenet/coexpressionnetwork. In these plots, the nodes represent genes and their colors represent the module assignment. Edges represent the adjacency between nodes and the edge color is a combination of the origin and target node colors. Due to the overwhelming number of edges in this network, the edge weights are scaled to minimize the range. Node size in this network is equivalent to the node's degree (e.g. the number of connections originating or terminating at that node) and the maximum node size is suppressed so as to provide maximal visual clarity.
2. Individual coexpression modules, as in *Figure 3F* and https://sites.google.com/a/g.ucla.edu/genenet/modules. These plots are similar to the preceding except that, potentially, more nodes are present in the module since the filtration procedures detailed above are applied in a different context (e.g. only the expression data in the module are considered here vs. the expression data for the entire network). The same scaling parameters as above are applied to the edges for visual clarity.
3. Protein interaction network, as in *Figure 7* and https://sites.google.com/a/g.ucla.edu/genenet/protein. Nodes represent proteins and their colors represent the coexpression module assignments. Node size is equivalent to its degree. Here, the edge width conveys meaning and is helpful in interpreting the relationship between nodes. An edge is drawn between two nodes when the STRING database indicates a high confidence interaction (score ≥0.9) between them. Edge widths are the confidence score scaled by the product of the origin and target node's intramodular connectivities (kIN). Thus, thick edges indicate a high confidence protein level interaction between two genes that are well connected members of learning and singing related modules. Unlike the previous plots, a node's size does not necessarily convey a higher degree of coexpression network importance. Instead, it indicates many interactions involving this protein described in the database. The thickness of the edges conveys influence of the gene's biological importance, as interpreted through their kIN. Whether a node's degree or the weight of its connections is the ultimate determinant of its relationship to vocal learning remains to be determined but the reader should keep the preceding information in mind when interpreting this network.

## Accession information

Raw and processed RNA-seq and behavioral data for each bird are available at the Gene Expression Omnibus (GEO; https://www.ncbi.nlm.nih.gov/geo/) at accession number GSE96843.

## Acknowledgements

We thank Jennifer Morales and Maria Truong for their assistance in analyzing song and non-vocal behavior. Insightful comments from three reviewers were provided on prior versions of the manuscript. This work was supported by NIH grant RO1MH07012 (SAW). ZDB received support from the 'Neuroendocrinology, Sex Differences, and Reproduction' training grant 5T32HD007228.

## Additional information

### Funding

| Funder | Grant reference number | Author |
| --- | --- | --- |
| National Institutes of Health | RO1MH070712 | Stephanie A White |
| National Institutes of Health | 5T32HD007228 | Zachary Daniel Burkett |
| Tennenbaum Center for the Biology of Creativity, University of California Los Angeles | | Stephanie A White |

The funders had no role in study design, data collection and interpretation, or the decision to submit the work for publication.

### Author contributions

Zachary Daniel Burkett, Conceptualization, Data curation, Formal analysis, Supervision, Funding acquisition, Investigation, Visualization, Methodology, Writing—original draft, Writing—review and editing; Nancy F Day, Conceptualization, Data curation, Formal analysis, Investigation, Writing—review and editing; Todd Haswell Kimball, Formal analysis, Investigation, Methodology; Caitlin M Aamodt, Validation, Investigation, Writing—review and editing; Jonathan B Heston, Conceptualization, Resources, Methodology, Writing—review and editing; Austin T Hilliard, Conceptualization, Data curation, Software, Methodology, Writing—review and editing; Xinshu Xiao, Conceptualization, Supervision, Methodology; Stephanie A White, Conceptualization, Resources, Supervision, Funding acquisition, Methodology, Writing—original draft, Writing—review and editing

### Author ORCIDs

Zachary Daniel Burkett http://orcid.org/0000-0002-5153-485X
Nancy F Day http://orcid.org/0000-0001-5367-9473
Jonathan B Heston http://orcid.org/0000-0001-7479-1122
Stephanie A White http://orcid.org/0000-0002-3490-2294

### Ethics

Animal experimentation: All animal use was in accordance with NIH guidelines for experiments involving vertebrate animals and approved by the University of California, Los Angeles Chancellor's Institutional Animal Care and Use Committee (IACUC) under protocol (#2001-54). All surgical procedures were performed under isoflurane anesthetic.

### Decision letter and Author response

Decision letter https://doi.org/10.7554/eLife.30649.055
Author response https://doi.org/10.7554/eLife.30649.056

## Additional files

### Supplementary files

• Supplementary file 1. Network data summary table. (A) xlsx table summarizing all gene level Area X and VSP network data for the juvenile birds presented in this study. Area X connectivity data in the adult network are included (*Hilliard et al., 2012a*). Columns are sortable using Microsoft Excel and defined as follows: [A] Gene symbol as annotated by NCBI *Taeniopygia guttata* genome assembly 3.2.4 annotation release 103. [B] Human homolog of the zebra finch genes in column A, as dictated by NCBI. [C] Gene's module assignment in the Area X overall network. NA implies the gene was in the VSP but not the Area X network. [D-E] Gene kIN and kTotal for juvenile Area X. [F-G] Gene kIN and kTotal for juvenile VSP. [H-I] Gene kIN and kTotal for adult Area X. [J-S] Gene significances ('gs') and q value for each gene-behavior trait relationship using Area X expression data. [T-BK] Module membership ('MM') as defined by correlation between the gene's expression profile across all samples and a module's eigengene and p-value for the correlation. [BL] Gene's module assignment in the VSP network. [BM-BT] Same as columns J through S except using VSP expression data.

DOI: https://doi.org/10.7554/eLife.30649.035

• Supplementary file 2. GeneAnalytics gene ontology information (A) xlsx workbook containing gene ontology information from the GeneCards GeneAnalytics module. For each module, there are four tabs sorted by default by descending module significance. Columns within each module are defined as follows: (1) Diseases: [A] Enrichment score. [B] Enrichment term. [C] Disease category. [D] Total genes annotated with this term. [E] Genes in this module annotated with this term. [F] Human gene symbols for genes in column E. [G] Link to disease page in GeneCards database. [H] Genetic associations (if applicable). [I] Matched genes for genetic associations (if applicable). [J] Differentially expressed in diseased tissues. [K] Matched genes for differential expression in diseased tissues. [L-P] Module and term significances for all behavior metrics (see Materials and methods). (2) GO – Biological Process, GO – Molecular Function, Pathways: [A] Enrichment score. [B] Enrichment term or pathway name. [C] Total genes annotated with this term or pathway. [D-E] Number and symbol of genes in this module annotated with this term. [F] Link to AMIGO or PathCards site for ontology term or pathway. [G-K] Module and term significances for all behavior metrics (see Materials and methods).

DOI: https://doi.org/10.7554/eLife.30649.036

• Supplementary file 3. Area X to VSP module preservation statistics Module preservation statistics with Z scores and Bonferroni-corrected log10 p-values are presented in columns C-AL. Column C is the basis for the scatter plot presented as *Figure 4B*.

DOI: https://doi.org/10.7554/eLife.30649.037

• Supplementary file 4. Juvenile Area X to adult Area X module preservation statistics Module preservation statistics with Z scores and Bonferroni-corrected log10 p-values are presented in columns C-AL. Column C is the basis for the scatter plot presented as *Figure 5B*.

DOI: https://doi.org/10.7554/eLife.30649.038

• Supplementary file 5. STRING database protein interaction data Pairwise protein level interactions between genes in singing and learning modules. These nodes were linked with edges and presented as *Figure 7*. Columns are defined as follows: [A-B] Nodes 1 and 2, one of which is in a song production module and the other in a learning module. [C-F] Methods for weighting the protein interaction scores as described in Methods: Protein Interaction Networks and Scaling of Interaction Confidence Scores. *Figure 7* uses weighted kIN (column C) as the metric for dictating edge width.

DOI: https://doi.org/10.7554/eLife.30649.039

• Transparent reporting form

DOI: https://doi.org/10.7554/eLife.30649.040

### Major datasets

The following dataset was generated:

| Author(s) | Year | Dataset title | Dataset URL | Database, license, and accessibility information |
|---|---|---|---|---|
| Burkett ZD, Day NF, Kimball TH, Aamodt CM, Heston JB, Hilliard AT, Xiao X, White SA | 2017 | Weighted gene coexpression analysis of RNA-seq data from 65d juvenile Area X and adjacent non-song ventral striatopallidum (VSP). | https://www.ncbi.nlm.nih.gov/geo/query/acc.cgi?acc=GSE96843 | Publicly available at the NCBI Gene Expression Omnibus (accession no: GSE96843) |

The following previously published dataset was used:

| Author(s) | Year | Dataset title | Dataset URL | Database, license, and accessibility information |
|---|---|---|---|---|
| Hilliard AT, Miller JE, Fraley E, Horvath S, White SA | 2012 | Weighted gene co-expression network analysis on microarray data from subregions of zebra finch (Taeniopygia guttata) basal ganglia | https://www.ncbi.nlm.nih.gov/geo/query/acc.cgi?acc=GSE34819 | Publicly available at the NCBI Gene Expression Omnibus (accession no: GSE34819) |

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
