## [Decision Letter]

[Editors’ note: a previous version of this study was rejected after peer review, but the authors submitted for reconsideration. The first decision letter after peer review is shown below.]

Thank you for submitting your work entitled "FoxP2 isoforms delineate spatiotemporal transcriptional networks for vocal learning in the zebra finch" for consideration by *eLife*. Your article has been reviewed by two peer reviewers, and the evaluation has been overseen by a Reviewing Editor and a Senior Editor. The reviewers have opted to remain anonymous.

Our decision has been reached after consultation between the reviewers. Based on these discussions and the individual reviews below, we regret to inform you that your work will not be considered further for publication in *eLife*.

As you see from detailed reviews, both reviewers appreciate your study, but raised substantial critiques that will take more than two months of work to address. As you may know, *eLife*'s policy of inviting a revision is that the authors should be able to return the revised manuscript that satisfactorily address reviewers' comments in about two months. We would be happy to consider a new submission of your manuscript in the future, should you be able to address these critiques. In addition to the initial reviews appended below, the reviewers provided additional guidance for improvement of the study, which I copy below:

The authors might consider one of the following trajectories:

a) Carrying out additional experiments to test whether any of the 'learning' networks normally are engaged during learning. For example, testing whether the levels of network expression co-vary with learning in unmanipulated birds. This is a significant enterprise that seems like it falls outside of the 'experiments that could be done in 2 months' criterion. However, it has the potential to show that networks identified in Area X and VSP through FOXP2 manipulation are actually relevant to normal developmental song learning. As I understand the authors' claims, a central prediction is that the 'learning networks' that they have identified are indeed relevant to normal function. I therefore think that this trajectory has the greatest likelihood of further substantiating the author's current conclusions. I think the results of such experiments are genuinely uncertain but, would not be surprised if the 'learning' networks identified in the current manuscript are not identified in an analysis of gene expression changes during normal learning (as opposed to being a signature for circuit damage/dysfunction driven by FOXP2 overexpression).

b) Reorienting their manuscript to focus more on what genes within and outside of learning circuitry are regulated/misregulated by FOXP2 isoforms, and potentially provide further insight into the regulatory action of FOXP2, and mechanistic insights into how this system contributes at a molecular level to organization of neural circuitry. In this orientation, abstract and manuscript would focus more on what does overexpression of different FOXP2 isoforms do, in terms of driving identifiable signatures in expression levels of downstream genes, as well as changes in gene networks. Per my thinking, these are potentially signatures of abnormalities associated with manipulations (or natural occurring mutations/misregulations) of FOXP2 function, and therefore have potential interest in shedding further insight into the mechanisms of FOXP2 regulation of gene expression and circuit function, independently of an ability to identify networks that are (for example) normally engaged during learning. I suspect that the authors might be able to reformulate their paper in this fashion with little or no additional data, and while I am open to other possibilities, it seems likely to me that there is a straighter conceptual line form the authors' experiments to this sort of formulation than to their current formulation. The one uncertainty I have about the authors' ability to reorient their paper in this fashion is the degree to which they can provide data from the experimental birds about the efficacy of viral manipulations (i.e. level of FOXP2 isoforms in each of their samples from manipulated and control birds). It isn't at all clear to me why the authors do not already have these data in hand, and what would impede an analysis of whether and how casual manipulations of FOXP2 isoforms drive changes to gene expression levels and gene networks. Indeed, some examination of these sorts of data regarding the efficacy and consequences of the experimental manipulations seems crucial whatever avenue the authors pursue. While it seems possible that the authors could follow this reformulation of their study with no additional experiments, it would reflect a sufficiently different approach from their current manuscript that it is hard to predict whether the authors could be successful.

Reviewer #1:

Previous work from this group and others has provided good evidence that Foxp2 plays a role in regulating both song learning in juvenile songbirds as well as song variability in adults. Considerable effort has been made in the field to dissect the molecular mechanisms by which Foxp2, particularly in the striatopallidal song nucleus Area X, influences song. This paper addresses two main questions: (1) How does the overexpression of two Foxp2 isoforms in Area X influence song learning and acute song features. (2) What gene co-expression patterns correlate with different singing and learning-related states, as generated by Foxp2 overexpression.

The authors' approach consisted of overexpressing three constructs via AAV injection into Area X of juvenile males: full-length Foxp2 (FoxP2.FL), a truncated naturally occurring isoform (FoxP2.10+) that lacks a DNA-binding domain, and a GFP control. These injections took place at 30 days post hatch, within the sensory and sensorimotor song learning periods. After 30 days song learning was assessed as tutor song similarity, and song variability was calculated as either song motif similarity across renditions or the change in the variability of song features (pitch, amplitude, etc.) after periods of free singing or sessions in which the bird was prevented from singing. The two Foxp2 isoforms were found to have separable effects on song learning and variability, with FoxP2.FL disrupting song learning and limiting song variability, and FoxP2.10+ having little effect on song learning but causing the characteristic song variability effect to shift in the opposite direction (i.e. instead of increasing after earlier undirected singing, variability decreased).

The authors extracted RNA from Area X and a control ventral striatopallidal (VSP) region of these birds, constructed and sequenced RNA-seq libraries from this RNA, then performed an extensive correlation-based gene expression network analysis using WGCNA. Consistent with previous work, they found several co-expression modules in Area X that correlate with amount of singing before euthanasia that they here term 'singing-related' modules, or sometimes 'song modules'. They also identify a few modules that correlate with the tutor song similarity termed 'learning-related' modules. Singing-related modules were not found independently in the VSP network or preserved from the Area X network in the VSP network. However, learning-related modules were preserved in VSP (but not found independently). Comparison to microarray Area X and VSP expression datasets from a previous study indicated that three of the song-related modules were preserved between juveniles and adults. In contrast, the authors find little preservation of the juvenile learning-related modules in adults. Finally, MAPK11, a Foxp2 target identified in mouse Foxp2 knockouts, is noted as the gene whose expression most strongly correlates with learning state and identification of a Foxp2 binding site in the promoter of zebra finch MAPK11 is given as additional evidence that this gene is a Foxp2 target in zebra finch.

As a whole, this work is an interesting look at how the genetic perturbation of different Foxp2 isoforms in the song system drives changes to song learning and variability. In addition, the RNA-seq dataset appears to be of high quality and offers an important glimpse into the expression patterns present in juvenile Area X and VSP under different genetic and behavioral states. The authors have also performed a great service in organizing this complex dataset and their network analyses into a readily accessible online format.

However, I have several major concerns about the analysis and especially the interpretation. The greatest issue is the interpretation of modules as 'learning related'.

I) There are a couple of ways to interpret the data showing correlations between a 'learning eigengene' and behavioral measures of learning:

1) Observed expression differences are due to direct action of Foxp2 isoform overexpression.

2) Expression differences are due to secondary effects from behavioral changes induced by Foxp2 OE.

It's the second interpretation the authors seem to rely on, using the molecular perturbation as a mechanism to generate behavioral diversity, possibly driving the system to extremes to better reveal underlying singing- and learning-related processes. This approach is potentially a powerful way to reveal mechanistic connections that may be undetectable by relying solely on natural variation in quality of learning and modulation of song variability. But it makes interpretability of a lot of the data difficult. For instance, assigning the label 'learning-related' to a given module seems uncertain given the learning-related effects are secondary to the manipulation. One could imagine a scenario in which FOXP2 overexpression drives some process that generally disrupts the functioning of Area X (for example a gross alteration in neuronal functioning, alteration in wiring, increase in apoptosis, etc.). Then we expect that learning will be impaired in proportion to disruption of Area X (since it is previously well established that damage of Area X impairs learning) and gene network analysis should detect networks associated with disruption and damage that do not necessarily relate to normal mechanisms of learning (such as apoptosis, microglia infiltration, axonal pruning, etc.). It would be incorrect in this scenario to conclude that the gene networks identified in this analysis are 'learning modules' rather than 'damage modules'. This seems to me to be a fundamental issue of interpretation for this study that significantly undermines the conclusions that the authors wish to draw.

Similarly, in the comparison between juveniles and adults, there is the argument the learning-related modules are present in both Area X and VSP in juveniles but in neither area of adults. However, in addition to age, a major difference between the groups is simply that juveniles have overexpression of Foxp2 isoforms while the adults are unmanipulated. This suggests that the observed effects might not be age-dependent but simply Foxp2 manipulated versus Foxp2 unmanipulated. The determination of whether FOXp2 isoforms are altered in VSP (as requested below), perhaps as a direct consequence of virus spillover into VSP, would aid in interpretation of these effects.

One compelling way to address the concerns expressed above would be to test whether the 'learning modules' identified under conditions of FOXP2 overexpression bear any resemblance to modules that are engaged during normal developmental learning. For example, within the GFP experimental group (or other control populations) there can be substantial variation in the quality of learning. If the authors could show that the learning modules identified in this study correlate with the quality of learning in control populations, that would go a long way towards substantiating the identity of these modules as learning related. Rather than carry out a full network analysis on additional birds that might be required to validate learning modules in control populations, the authors could alternatively analyze expression levels of a set of hub genes identified as part of the learning module.

More broadly, further analysis of the nature of expression changes driven by FOXP2 overexpression would be helpful in interpreting data.

a) It would be helpful to show a plot demonstrating the differential expression of FoxP2 in the overexpression conditions relative to the control GFP condition. Particularly informative would be separation of the two isoforms using reads mapped to distinguishing exons. This should be done for both Area X and VSP datasets to get a sense of how much overexpression there is in VSP. This point seems particularly important as the data from these analyses should reveal the effectiveness of the causal manipulations in the study, and aid in interpretation of learning effects and the links between modules and behavioral traits.

b) Also required is some form of differential expression analysis comparing Area X and VSP as well as Area X control versus overexpression conditions. Correlation-based network analysis can be an extremely effective tool in reducing the complexity of large expression datasets (as nicely shown here). But its use does not supplant the need for some basic understanding of what FoxP2 overexpression does to gene expression levels. This requirement is particularly germane with the manipulation of a transcription factor, which, if effective, should have direct effects on expression. Regarding the interpretation that there are learning modules in juvenile VSP and Area X that are absent in adults, it is especially important to establish whether viral infection has driven changes in FOXP2 expression in VSP, perhaps due to direct viral infection of that region. If FoxP2 overexpression is driven in VSP (as well as Area X) this would further accentuate concern that the 'learning module' detected in both VSP and Area X reflects direct consequences of overexpression in both of these regions, rather than processes in these regions that are specifically linked to learning.

c) For gene expression data it would be helpful to provide a plot showing sample distance determined from the expression data to give an indication of how well the replicate datasets are clustering. This could take the form of some sort of dimensionality reduction approach such MDS, PCA, or tSNE.

II) The analyses are complicated enough that I find it difficult to assess the validity of conclusions. While more analyses, presentation of statistics, and discussion has the potential to be persuasive, the most compelling way to validate the conclusions from the analysis of the FOXP2 manipulated birds is to show that the conclusions derived from the analyses accurately predict or inform findings in other settings. As outlined above, most compelling would be demonstration that the 'learning modules' identified under conditions of overexpression are also relevant to the normal juvenile song learning process. The MAPK11 analyses are also a start in the direction of demonstrating the utility of the network analysis, and if further supported would provide another way of validating the network structures identified in the study.

The authors find that MAPK11 expression is most strongly correlated with learning and is a Foxp2 target as determined in a mouse knockout study. However, this conclusion needs further support: identification of a single TFBS in a promoter can potentially not be that meaningful. For the analysis to be stronger, the following questions should be addressed:

What is the probability that this motif was found randomly?

Is the Foxp2 motif enriched in MAPK11 connected genes or learning-associated modules?

If you perform a standard differential expression analysis between control and Foxp2 conditions, do you find a consistent set of misregulated genes? Are these gene promoters enriched for Foxp2 motifs?

Additionally, the drawing of connections between MAPK11, ATF2, and H2B needs further support to be persuasive. This is especially apparent in the reasoning concerning H2B. Histone H2B is a core nucleosome component, constitutively expressed across tissues. Its post-translational modification has important roles in regulating transcription, but it's unclear if regulation of its expression per se has roles in modifying the expression of specific genes. The argument here is that H2B expression levels are modulated in response to singing due to the gene's inclusion in the blue network. The authors include a citation in support of this argument. However, the paper cited describes H2B acetylation differences in a learning paradigm and not expression differences, which would be the more direct comparison.

*Reviewer #2:*

This study examines the relationships between striatal expression of FOXP2 isoforms and vocal learning and variability in a songbird. This is an important contribution, in particular because this gene's involvement in vocal learning cannot be properly studied in models like mice, which lack this behavior. The study takes advantage of the knowledge on vocal learning and circuitry in finches and makes extensive use of weighted co-expression analysis across ages and basal ganglia subregions. It contributes novel data and insights, indicating that different gene sets are co-expressed in correlation with different states of singing or learning. It also points to novel relationships among genes, and candidates for mechanistic studies. The effort is a tour de force, the expression data are impressive, and the findings seem robust.

One issue is that the study does not provide direct evidence that the major variant examined, the short 10+ isoform, occurs endogenously in finch brain tissue. The fact that this variant transcript exists is not sufficient, as transcripts do not reliably reflect expression at protein level. Because this point is novel and central, the authors should consider options for demonstrating the endogenous occurrence of this protein isoform, such as mass spectrometry to detect the isoform's unique peptide or developing a specific antibody. The lack of direct evidence weakens a major underlying assumption of the study and will require toning down several statements about expression of that isoform and the implications for many of the datasets presented.

With regards to MAPK11, the authors' inference about the putative transcription start site (TSS) and promoter definition is likely incorrect. In silico predictions are often incomplete and need to be taken with caution in the absence of data like cDNAs with intact 5' caps, or primer extension assays. The orthologous MAPK11 RefSeq models from other species (chicken, mammals, *Xenopus*) do not map to the zebra finch genome for their first 100-200 nucleotides (readily verified with alignments in UCSC's genome browser). In fact, this 5' segment in other species corresponds to a first exon that maps several kb upstream of the second exon for this gene in the respective genomes (~12 kb in chicken, where the gene has 12 exons). Assuming the study was based on Ensembl's prediction in zebra finch (please indicate if otherwise), this first exon is not predicted: the model has only 11 exons and does not have a starting ATG, indicating 5' end incompleteness. A 5'-most exon, if found, would bring the exon number to 12 and likely lie several kb upstream (possibly in a gap) to the first exon of the Ensembl model. Thus, the authors have most likely not identified the correct TSS/promoter region for this gene in finch. Further, to assess the significance of the occurrence of any given DNA binding motif, an estimate of chance occurrences in the genome needs to be provided. In sum, the claims about the presence of a possible FOXP2 binding site(s) in promoter region of MAPK11 are not supported by the evidence provided.

[Editors’ note: what now follows is the decision letter after the authors submitted for further consideration.]

Thank you for submitting your article "FoxP2 isoforms delineate spatiotemporal transcriptional networks for vocal learning in the zebra finch" for consideration by *eLife*. Your article has been reviewed by three peer reviewers, and the evaluation has been overseen by a Reviewing Editor and a Senior Editor. The following individuals involved in review of your submission have agreed to reveal their identity: Erich D Jarvis (Reviewer #2).

The reviewers have discussed the reviews with one another and the Reviewing Editor has drafted this decision to help you prepare a revised submission.

We apologize for the time it took to get back to you. We have now received three reports: two original reviewers and one new reviewer. As you see from their detailed reviews, while they are in general supportive of this study, there are substantial remaining issues that need to be thoroughly addressed before the manuscript is acceptable for publication in *eLife*. We want to emphasize that if there are substantial issues remained in a further revised manuscript, it will have to be rejected. Please take your time to do a thorough revision addressing as much as possible reviewers' critiques. Below is a list of the most important points, consensus from extensive discussions among the reviewers after they have all submitted their individual reviews.

1) As pointed out by reviewer 2 (the new reviewer), echoed by previous rounds of both reviewers 1 and 3 and remaining concern of reviewer 3, a major issue is whether your data provide unequivocal support to the notion that FoxP2 manipulations affect area X gene expression and gene network (a major claim of the manuscript that is reflected in the title). This 'learning network' is present in both control (GFP) and all animals combined and is generally correlated with tutor song similarity across groups, but there is no evidence that it or any other network was affected by the FoxP2 manipulations. Short of asking you to redo the experiment of analyzing gene expression in only virally transduced regions, you are strongly encouraged to dig further into the analyses comparing specific genes, including more qRT-PCR on control genes to find that maybe there is a viral FoxP2 manipulated gene expression signal in the Area X punches. If you cannot find gene expression changes due to FoxP2 manipulations, you should state clearly the results, substantially modify this part of the manuscript, including the title.

2) All reviewers expressed frustrations that much of the data and analyses you described in the rebuttal did not make into the revised manuscript. Please include useful materials in revised manuscript, at least in supplemental figures, so that readers can have additional perspectives.

3) You need to spell out more clearly in the revision (not just the rebuttal) where you did/did not correct for multiple comparisons and rationalization for including 'non-significant' correlations in further analysis.

4) Reviewer 3 has substantial concern whether the data allow you to claim that FoxP2 binds to in the MAPK11 promoter rather than intron. Calling it a regulatory region might be more appropriate.

Reviewer #1:

The authors have revised the manuscript substantially, addressing several of the main issues that were previously raised. This includes a re-analysis of the 'learning-associated' module in control birds, as well as further data showing the endogenous expression of the truncated FOXP2 protein isoform and providing evidence for binding of FOXP2 binding to a predicted site in the MAPK11 candidate target gene. The authors also provided clarifications on several other important methodological and interpretation points. The manuscript is thus much stronger and better supported by the data.

Some concerns remain, however. The authors have not directly addressed the multiple comparisons and statistical significance. They acknowledge the issue in the rebuttal, but apparently have not made significant related adjustments in the paper, and still show all the correlations performed. Arguably the study lacked statistical power to perform all these multiple comparisons and correlations, given the sample size and the scope of the effort. The authors state, though, that some correlations were stronger and remained significant even after FDR corrections. Thus, perhaps a reasonable compromise would be to indicate which correlations survived the FDR correction, and suggest that others might be significant with a larger n and/or different design.

The effort to demonstrate FOXP2 binding to a predicted genome region is a very significant experimental result and considerably strengthens the paper, providing a plausible link between FOXP2 protein and MAPK11 as a likely target. However, the concern remains whether the authors have identified the 'promoter' region for MAPK11. As detailed below, some important lines of evidence question that assumption, as can be verified by examining genome alignments on a browser like UCSC's:

1) There is no expression evidence (cloned mRNAs/cDNAs with intact 5'UTRs, RNAseq maps, or data from a primer extension assay or equivalent), to verify the transcription start site (TSS) for this gene. Thus, the identification of the putative TSS relies solely on the position of the first exon predicted by the RefSeq model. This is worrisome as there are several gaps in the zebra finch genome upstream of the identified MAPK11 exons, and no confirmation that this RefSeq is complete.

2) The Ensembl model for zebra finch MAPK11 is longer than the RefSeq (Ensembl: 323 vs RefSeq: 285 residues), due to a longer N-terminus region, but it still does not contain a predicted starting ATG, further indicating that the RefSeq prediction for zebra finch MAPK11 is incomplete at the 5' end.

3) The chicken MAPK11 RefSeq (NM_001006227.1) is considerably longer than the zebra finch RefSeq (361 vs 285 predicted residues) and contains a predicted N-terminus peptide (MSERGGFYRQELNKTVWEVPQRYQNLTPVGSGAYGSVC) that maps to the chicken genome at ~12kb upstream of the second exon of chicken MAPK11. This peptide and corresponding exon position are well supported by chicken cloned mRNAs (e.g. AJ720776). This peptide is missing from the zebra finch MAPK11 RefSeq and does not map to the zebra finch genome, likely because of the gaps located ~13kb upstream of the first identified exon of zebra finch MAPK11 as predicted by Refseq or Ensembl (3 gaps between bp 19,659,000 and 19,665,000 on chr1A).

4) While the lack of alignment of the first chicken exon to the finch genome could be due to low conservation, the predicted N-terminus peptides of MAPK11 in at least 5 other songbird species (the closely related Bengalese finch, as well as starling, white-throated sparrow, great tit and Tibetan ground-tit) are practically identical to that in chicken, and similarly do not align to the zebra finch genome. However, the same N-terminus peptides in all these 5 songbird species align well to the chicken genome at the exact location of the 1st exon of chicken MAPK11. Further, this same N-terminus peptide from these birds is also present in MAPK11 in mouse and humans and aligns well at the position of the first exon, indicating its high conservation. This further strengthens the conclusion that the zebra finch RefSeq is incomplete at the 5' end and does not contain the first exon for MAPK11, and thus cannot be relied on to define the TSS for this gene in this species.

Rather than having identified the TSS and thus the promoter region for zebra finch MAPK11, it is much more likely that the authors' data provide evidence for FOXP2 binding at an internal, presumably intronic site. This is quite interesting in itself, and consistent with data from other genes and organisms that relevant transcription factor binding sites are not restricted to promoter regions. The author's interpretation that they have identified the promoter region for MAPK11 is less tenable without further data, as it would imply that this gene in zebra finches has a considerable truncation at its 5'-region, resulting in the loss of a N-terminus peptide that is conserved across vertebrates, including other closely related songbirds and mammals.

These issues should be acknowledged and the claim that the TSS/promoter has been identified needs to be toned down. It would also be helpful if the authors could deposit the sequence(s) identified in the ChIP assay for FOXP2 in GenBank, to allow for precise mapping onto the zebra finch genome.

Reviewer #2:

I was asked to review the original version of this paper but could not at the time due to time constraints. However, I reviewed the revised version, and did so first without looking at the original version or the reviewer's comments on that version. So this represents a fresh view of the revised paper. I only looked over the prior reviews and reviewer comments afterwards.

I think the underlying question, experiments, and raw data generated in this manuscript is very good. This includes finding that 2 different isoforms of FoxP2 have different functions in the striatal song nucleus AreaX for song production and song learning, and that there are differences as well as overlaps in the song production and song learning gene regulatory networks of juveniles versus adults.

However, I see issues with the preparation of the paper, interpretations, and one of the important experiments on viral manipulation of gene expression as measured by RNASeq in Area X. For preparation, the manuscript is not presented clearly enough, causing confusion to the reader. Sometimes discussion is in the results and vice versa; figures are not explained well enough. For interpretations, an example is the words 'learning' and 'song', which are used too loosely, as opposed to 'song learning' and 'song production'. Another is the function of the VSP, with is mostly a motor area, but not explained. There are more of these examples highlighted in my specific comments below. The most worrying item is the RNASeq FoxP2 manipulation analyses. The authors skipped a step in the analyses in the paper, which is determining the differences of FoxP2 manipulation in the basal- and singing-driven gene expression of Area X. It looks like this might not be possible to do with their current data, because they punched biopsied of the entire Area X, and not the 25% of it that had the gene manipulation. They might have been able to see that 25% if they used a GFP marker under a fluorescent scope in their dissections of the tissue. What they did instead was group all the samples together (control and manipulated animals) and created a network expression analyses in Figure 3, with the assumption that it is affected by the viral vector manipulations in Area X. One would hope that there is a partial affect from the 25% of cells of the manipulated Area X on the RNASeq data, and this can be tested. It took me some effort in re-reading to figure out what the authors did.

After reading the reviews, I see original reviewers 1 and 2 had the same concern, just maybe not expressed as clearly. In response to that concern, the authors did a GFP manipulated Area X only analyses and came up with the same results (Author response image 1 of the response letter). This is good and should be included in the main paper. However, they did not show network analyses of the viral FoxP2 manipulated animals, but instead show that clustering of expression does not differ between the GFP and FoxP2 manipulated Area X (Author response image 7 of response letter). So, if we take their analyses at face value, this would mean that the combined network analyses of control and FoxP2 manipulated animals in Figure 3 of the manuscript may not be influenced by the FoxP2 manipulation, and instead is simply a control network similar to unmanipulated animals. This needs to be shown. Further, if it is not affected by the FoxP2 manipulations, then it is a flaw of the experimental design of the punch biopsy approach and the sentence, in subsection the “Gene modules in juvenile Area X that correlate to vocal behavior were enriched for communication and intellectual disability risk genes”: “We used RNA-seq to quantify gene transcription in Area X of 65d juveniles overexpressing GFP, FoxP2.FL or FoxP2.10+, then used WGCNA to identify gene co-expression modules and link them to song learning”). If a lack of a difference in control and FoxP2 manipulated animals is true, then I think the RNASeq experiments should be presented in another paper, because the potential FoxP2 manipulation would have been drowned out and this is still an interesting finding in the normal singing juvenile vs adult states. However, if a small affect is found in the noise of the 75% unmanipulated Area X, then that affect would belong in this paper.

Reviewer #3:

The authors have been very responsive in their revisions and have made several key additions and clarifications that substantially improve the manuscript. Specific comments are included below regarding these main points. The residual issues raised below are ones that the authors should be able to address in Discussion section or with modest additional work.

1) Particularly nice is that they have found a correlation between the "learning network" and tutor song similarity for the GFP control group of birds. This significantly addresses one of my primary concerns in the initial submission regarding whether the network that was identified in FOXP2 manipulated birds would be present and correlated with learning in a control population. It is impressive that the green module was recovered in the GFP-only samples and that its correlation with tutor similarity was maintained. I find this result particularly interesting in conjunction with the analysis in Figure 6, showing that MAPK11 levels perfectly correlate with tutor similarity within virus conditions.

2) An important related addition is the PCR analysis of FOXP2 expression levels in ventral striatum. These data further support the authors contention that the presence of the learning network in both Area X and ventral striatum is not a direct result of the viral manipulations (spillover to ventral striatum) in both nuclei.

3) The further elaboration regarding MAPK11-ATF2-H2B potential link to learning is also a good addition that is more clearly grounded in previous work and seems less speculative.

4) Related to 3: the intent behind the FOXP2 ChIP experiments is excellent, and such data could potentially provide further support the authors' conclusions. However, to be compelling they would require two additions:

a) Presentation of qPCR instead of (non-quantitative) endpoint PCR data.b) Inclusion of at least 2 negative control regions to enhance the evidence for specificity (The IgG control is good but does not control for within-antibody noisiness).

I think the authors ought to be able to provide these additional data without too much additional work. If, however, this is problematic, an alternative would be to exclude these data and instead expand the bioinformatic analysis as suggested in the previous round of reviews to assess the statistical significance of the FOXP2 motif upstream of the MAPK11 TSS (I.e. to judge how likely this is by chance.)

5) The clustering analysis of the samples by their expression profiles is good and should be included in the text.

6) General Interpretation. The revised manuscript more firmly establishes an association between the 'green network' and tutor song similarity. This association rationalizes the use of 'learning network' to refer to the green network. However, the authors should be clear in their discussion of this association that these data are primarily correlative and the role of this network in learning per se remains speculative and a subject for future investigation. I think that the authors intend to make the point that the presence of this network in the VSP indicates that it may capture more general (not song specific) aspects of nervous system development or plasticity that correlate with song learning; if so, this could be made more explicit in Discussion section. Along those lines, it seems that such generalized changes could arise (and be correlated with tutor song similarity) in a variety of ways, which could reflect more or less direct involvement of this pathway in song learning. For example, both song learning and general ns development (captured by green network) might depend on a third variable (hormonal status, variation in neuromodulatory tone, social interactions and stimulation associated with effective tutor song experience, etc.). Indeed, it even seems possible that tutor song similarity achieved by different individuals itself feeds back to influence social and hormonal status of birds in ways that promote or inhibit more general aspects of nervous system development, or plasticity networks that could be captured by the green network. The authors need not belabor these points, but I think they would be well served to at least briefly discuss alternatives to the possibility that variation in the green network is tightly linked to, and is causally driving, variation in song learning (which is how the treatment currently reads to me). In this context, the mechanistic investigations of network identity/membership/function (e.g. MAPK11-ATF2-H2B) are nice in that they illustrate how candidate mechanisms can be identified through network analysis and begin to build a stronger case for specific mechanisms.

7) In general the authors addressed many issues in reasonable fashion in the rebuttal but for many of them did not address these points in the revision. In some cases the authors note that the requested analyses were not included in the revision because 'the results were negative', but in these cases a negative result is informative and effectively addresses potential concerns about interpretation of the primary data. Where these data (in rebuttal) address concerns raised by the reviewers they are also likely to be of interest to readers of the published manuscript, and as much as possible the authors would be well served to include most of these analyses in the revised manuscript – perhaps in supplementary material.

---

## [Author Response]

[Editors’ note: the author responses to the first round of peer review follow.]

The authors might consider one of the following trajectories:a) Carrying out additional experiments to test whether any of the 'learning' networks normally are engaged during learning. For example, testing whether the levels of network expression co-vary with learning in unmanipulated birds. This is a significant enterprise that seems like it falls outside of the 'experiments that could be done in 2 months' criterion. However, it has the potential to show that networks identified in Area X and VSP through FOXP2 manipulation are actually relevant to normal developmental song learning. As I understand the authors' claims, a central prediction is that the 'learning networks' that they have identified are indeed relevant to normal function. I therefore think that this trajectory has the greatest likelihood of further substantiating the author's current conclusions. I think the results of such experiments are genuinely uncertain but would not be surprised if the 'learning' networks identified in the current manuscript are not identified in an analysis of gene expression changes during normal learning (as opposed to being a signature for circuit damage/dysfunction driven by FOXP2 overexpression).

As suggested by both the editors and reviewers, we were able to use the existing data set to address ‘trajectory A’ and found that the ‘learning’ networks are indeed engaged during normal learning in our control animals. To do so, we constructed a gene co-expression network from only the 7 GFP control samples that made up a portion of the network presented in the manuscript (herein referred to as the ‘all-samples network; See Author response image 1, below), then performed differential network analysis between the two networks. Our hypotheses were as follows:

1) If the green learning module is the product of FoxP2 overexpression, drastically different connectivity of its genes and poor overall module preservation would be observed in the GFP-only network.

2) If the green learning module is related to learning as a product of FoxP2 overexpression, the gene significance of the green module eigengene (if the module exists at all) would be poorly correlated to learning.

In our analysis of the GFP-only network, we observed:

1) All of the modules, including the green module, are very well preserved between the GFP-only network and the all-samples network. (See Author response image 2, below).

2) The green learning module eigengene is more strongly correlated to vocal learning outcome in the GFP-only network than it is in the all-samples network. This correlation produces a p-value that passes false discovery rate correction. (See Author response image 3, below).

**Author response image 1. respfig1:** GFP-only network dendrogram, modules, and gene significances for investigated traits. Module colors are preserved with the all-samples network modules (e.g. main text Figure 3) if significant overlap between modules was observed between the two networks. Like the all-samples network, the green module here displays a strong positive correlation to learning even though gene significance is computed only from GFP-injected birds’ learning data.

**Author response image 2. respfig2:** Module preservation statistics between all-samples network and GFP-only network. The green module is extremely well preserved between the two networks, as indicated by Zsummary (left) and preservation rank (right).

**Author response image 3. respfig3:** GFP-only network module eigengenes with Pearson correlations to analyzed behaviors and FDR-corrected p-values. Module colors are consistent between the GFP-only and all-samples networks when significant overlap between the modules exists.

Reviewer #1:As a whole, this work is an interesting look at how the genetic perturbation of different Foxp2 isoforms in the song system drives changes to song learning and variability. In addition, the RNA-seq dataset appears to be of high quality and offers an important glimpse into the expression patterns present in juvenile Area X and VSP under different genetic and behavioral states. The authors have also performed a great service in organizing this complex dataset and their network analyses into a readily accessible online format.

We appreciate reviewer 1's enthusiasm for our work, including that 'the RNA-seq data set appears to be of high quality and offers an important glimpse into the expression patterns present in juvenile Area X and VSP…' and that we have 'performed a great service' in organizing the data set.

However, I have several major concerns about the analysis and especially the interpretation. The greatest issue is the interpretation of modules as 'learning related'.I) There are a couple of ways to interpret the data showing correlations between a 'learning eigengene' and behavioral measures of learning:1) Observed expression differences are due to direct action of Foxp2 isoform overexpression.2) Expression differences are due to secondary effects from behavioral changes induced by Foxp2 OE.It's the second interpretation the authors seem to rely on, using the molecular perturbation as a mechanism to generate behavioral diversity, possibly driving the system to extremes to better reveal underlying singing- and learning-related processes. This approach is potentially a powerful way to reveal mechanistic connections that may be undetectable by relying solely on natural variation in quality of learning and modulation of song variability. But it makes interpretability of a lot of the data difficult. For instance, assigning the label 'learning-related' to a given module seems uncertain given the learning-related effects are secondary to the manipulation. One could imagine a scenario in which FOXP2 overexpression drives some process that generally disrupts the functioning of Area X (for example a gross alteration in neuronal functioning, alteration in wiring, increase in apoptosis, etc.). Then we expect that learning will be impaired in proportion to disruption of Area X (since it is previously well established that damage of Area X impairs learning) and gene network analysis should detect networks associated with disruption and damage that do not necessarily relate to normal mechanisms of learning (such as apoptosis, microglia infiltration, axonal pruning, etc.). It would be incorrect in this scenario to conclude that the gene networks identified in this analysis are 'learning modules' rather than 'damage modules'. This seems to me to be a fundamental issue of interpretation for this study that significantly undermines the conclusions that the authors wish to draw.

Again, we appreciate the reviewer's sentiment that our approach has the potential to be 'a powerful way to reveal mechanistic connections.' With regard to the second interpretation, that our intervention led to damage, we offer the following three broad arguments. First, we failed to point out that the behavioural changes seen here are vastly different than those induced by damage to Area X. As shown by Scharff and Nottebohm, 1991), damage to Area X leads to high sequence entropy and longer syllables, neither of which were observed here. Rather, like Haesler et al., (2010) who knocked down FoxP2, and Heston et al., (2015) in which we overexpressed FoxP2, we found incomplete syllable copying, poor syllable copying and feature specific errors. Moreover, birds in both this study, our prior one, and in Haesler et al. were still able to change their songs over development. They were just unable to match them to the tutor. We have added language to address this point in subsection “Virus-mediated overexpression of FoxP2 isoforms affected song learning and/or vocal variability” in the fourth paragraph and present below a learning trajectory curve (Author response image 4) where birds’ songs throughout the song learning period are compared to their ~65d song instead of the tutor’s. We opt not to include these data as a figure in the manuscript because the results are negative and the findings have been presented in the manuscripts mentioned above.

**Author response image 4. respfig4:** Learning trajectories demonstrate that birds’ songs change over song development. Learning trajectory is calculated by quantifying acoustic similarity between birds’ songs over the course of the experiment and their song on the day of sacrifice (~65d). Upward trajectory indicates the songs for all three virus groups change over time. In agreement with the work of Heston & White, (2015), the FoxP2.FL group’s (black) song changes over time but, as illustrated in main text Figure 3, not to match the tutor’s song.

Second, if disruption and damage had occurred, we might not have expected to recover the song modules, presented in main text Figure 5. As the reviewer notes below, song modules were originally identified in adult birds that were unmanipulated (Hilliard et al., 2012). Their preservation in our study, conducted using a different platform, years later, in different animals at a different age and by different experimentalists provides support that Area X was not grossly aberrant.

Third, the work of Heston & White, (2015) using identical viruses and injection volumes revealed no abnormal histology in the striatopallidum.

Similarly, in the comparison between juveniles and adults, there is the argument the learning-related modules are present in both Area X and VSP in juveniles but in neither area of adults. However, in addition to age, a major difference between the groups is simply that juveniles have overexpression of Foxp2 isoforms while the adults are unmanipulated. This suggests that the observed effects might not be age-dependent but simply Foxp2 manipulated versus Foxp2 unmanipulated. The determination of whether FOXp2 isoforms are altered in VSP (as requested below), perhaps as a direct consequence of virus spillover into VSP, would aid in interpretation of these effects.

We note the preservation of song module genes between manipulated and unmanipulated animals mentioned above. In response to the reviewer's request here and below, we investigated whether or not FoxP2 isoforms were upregulated in outlying VSP. Using the same qRT-PCR protocol that was sufficient to detect overexpression of the FoxP2 isoforms in Area X samples and presented in main text Figure 1, we observed no significant difference in FoxP2 level across virus groups with either primer set in the VSP samples (see Author response image 5, below). We have added language to this extent in the Results section. We opt not to present these data in a figure in the main text because the results are negative.

**Author response image 5. respfig5:** FoxP2 overexpression is not evident in VSP samples. Using the same primers and reaction conditions as the Area X samples (main text Figure 1), FoxP2 overexpression is not observed in either virus group relative to GFP.

To summarize, our new findings indicate that the green module is not simply the product of FoxP2 overexpression but rather reflects normal gene co-expression underlying vocal learning. The green module’s co-expression pattern exists in birds injected with only the GFP control virus construct and is strongly and positively correlated to learning behavior in those animals. The green module also exists in the VSP samples from all of the birds in the study, a region which does not show signs of FoxP2 overexpression.

One compelling way to address the concerns expressed above would be to test whether the 'learning modules' identified under conditions of FOXP2 overexpression bear any resemblance to modules that are engaged during normal developmental learning. For example, within the GFP experimental group (or other control populations) there can be substantial variation in the quality of learning. If the authors could show that the learning modules identified in this study correlate with the quality of learning in control populations, that would go a long way towards substantiating the identity of these modules as learning related. Rather than carry out a full network analysis on additional birds that might be required to validate learning modules in control populations, the authors could alternatively analyze expression levels of a set of hub genes identified as part of the learning module.

The presence of the learning modules in the GFP control birds, detailed above, addresses this concern.

More broadly, further analysis of the nature of expression changes driven by FOXP2 overexpression would be helpful in interpreting data.a) It would be helpful to show a plot demonstrating the differential expression of FoxP2 in the overexpression conditions relative to the control GFP condition. Particularly informative would be separation of the two isoforms using reads mapped to distinguishing exons. This should be done for both Area X and VSP datasets to get a sense of how much overexpression there is in VSP. This point seems particularly important as the data from these analyses should reveal the effectiveness of the causal manipulations in the study, and aid in interpretation of learning effects and the links between modules and behavioral traits.

With regard to differential FoxP2 overexpression, we refer to the reviewer to Figure 1 in which qRT-PCR was used to quantify FoxP2 transcript levels in Area X as a function of viral construct (i.e. GFP, FoxP2.FL, and FoxP2.10+) and compare to Author response image 5 (above) where no overexpression is evident in VSP.

The sequencing data alone do not demonstrate overexpression of the FoxP2 isoforms. Quantification of the cells infected at the injection site using the identical procedures to ours here (see Heston & White, 2015) indicates 20-25% of cells, the majority of which are FoxP2+ medium spiny neurons, are infected by the virus. Therefore, in our Area X samples, 70-75% of the medium spiny neurons are not overexpressing FoxP2. This presents a challenging situation: a clear and reproducible behavioral effect is observed with the virus manipulation, but the transcriptional basis for it is only visible in the qRT-PCR data presumably due to the amplification required to overcome a low signal-to-noise ratio.

b) Also required is some form of differential expression analysis comparing Area X and VSP as well as Area X control versus overexpression conditions. Correlation-based network analysis can be an extremely effective tool in reducing the complexity of large expression datasets (as nicely shown here). But its use does not supplant the need for some basic understanding of what FoxP2 overexpression does to gene expression levels. This requirement is particularly germane with the manipulation of a transcription factor, which, if effective, should have direct effects on expression. Regarding the interpretation that there are learning modules in juvenile VSP and Area X that are absent in adults, it is especially important to establish whether viral infection has driven changes in FOXP2 expression in VSP, perhaps due to direct viral infection of that region. If FoxP2 overexpression is driven in VSP (as well as Area X) this would further accentuate concern that the 'learning module' detected in both VSP and Area X reflects direct consequences of overexpression in both of these regions, rather than processes in these regions that are specifically linked to learning.

We respectfully disagree regarding the necessity to present detailed differential expression analyses between Area X and VSP. In this study and in our prior work (Hilliard et al., 2012), differential expression between Area X and VSP are presented directly adjacent to figures plotting differential connectivity (see main text Figure 4). Our data indicate that a gene’s regionality and/or relevance to behavior is captured more effectively by considering its transcriptional context (i.e. connectivity) through network analysis instead of through changes in absolute expression level. Moreover, our expression data will be made available through GEO upon the manuscript’s publication so that interested parties can conduct differential expression analyses.

As indicated above, we find no evidence of FoxP2 overexpression in VSP either by: (1) *in situ* analysis, (2) examination of RNA-seq data from VSP punches, or (3) now in response to the Reviewer, qRT-PCR (Author response image 5, above).

c) For gene expression data it would be helpful to provide a plot showing sample distance determined from the expression data to give an indication of how well the replicate datasets are clustering. This could take the form of some sort of dimensionality reduction approach such MDS, PCA, or tSNE.

In response to the reviewer’s request, we present figures below depicting how our samples relate to each other. When samples from both Area X and VSP are considered together, the samples clearly separate by brain region (see Author response image 6).

Within the Area X samples, grouping is not clearly delineated by viral construct (see Author response image 7). This is likely the product of a fairly subtle manipulation (<25% cellular transduction), noted above. Further, as described in this manuscript and our prior work (Hilliard et al., 2012), the act of singing has a dramatic effect on the transcriptome, correlating with changes in expression of thousands of genes. The animals in this study sang across a wide continuum (0 to ~1900 motifs) in the two hours immediately preceding sacrifice. Thus, our study does have biological replicates within the virus groups, but there are no replicates for singing since the animals were allowed to sing ad libitum before sacrifice. This variability is ideal for gene network analysis but creates the situation where a virus by singing interaction may occur, driving the clustering pattern we observe here.

**Author response image 6. respfig6:** The intersample correlation (y-axis) delineates a clear split between Area X samples (green) and VSP samples (red). One VSP sample (White383V) clustered with the Area X samples but was not included in the Area X network.

**Author response image 7. respfig7:** The intersample correlation for the Area X samples does not clearly delineate clusters by virus construct. Virus construct is indicated in green (GFP), black (FoxP2.FL), and red (FoxP2.10+). Below, time singing is indicated on a blue-red color scale where the samples from birds that sang the most are colored the deepest red.

II) The analyses are complicated enough that I find it difficult to assess the validity of conclusions. While more analyses, presentation of statistics, and discussion has the potential to be persuasive, the most compelling way to validate the conclusions from the analysis of the FOXP2 manipulated birds is to show that the conclusions derived from the analyses accurately predict or inform findings in other settings. As outlined above, most compelling would be demonstration that the 'learning modules' identified under conditions of overexpression are also relevant to the normal juvenile song learning process. The MAPK11 analyses are also a start in the direction of demonstrating the utility of the network analysis, and if further supported would provide another way of validating the network structures identified in the study.

We agree that the most compelling demonstration is to find the ‘learning modules’ in normal juvenile birds. Thanks to the reviewer’s point which was shared with the other reviews, this has now been demonstrated using birds injected with the GFP-expressing virus. Going further and as also requested, we present new data from chromatin immunoprecipitation experiments indicating that FoxP2 binds the MAPK11 promoter, providing a mechanistic basis for a prediction made by our network data. The results of these experiments have been added to the main text in the subsection “A bioinformatics approach indicates MAPK11 as an entry point to neuromolecular networks for vocal learning” and in Figure 6. The methods for these experiments have been added to the Materials and methods section of the manuscript. It is our hope that by publishing the manuscript we can share the dataset with the scientific community enabling additional follow up on the highlighted pathways.

The authors find that MAPK11 expression is most strongly correlated with learning and is a Foxp2 target as determined in a mouse knockout study. However, this conclusion needs further support: identification of a single TFBS in a promoter can potentially not be that meaningful. For the analysis to be stronger, the following questions should be addressed:What is the probability that this motif was found randomly?Is the Foxp2 motif enriched in MAPK11 connected genes or learning-associated modules?If you perform a standard differential expression analysis between control and Foxp2 conditions, do you find a consistent set of misregulated genes? Are these gene promoters enriched for Foxp2 motifs?

In response to the reviewer's queries, we conducted a direct functional test of the predicted interaction between FoxP2 and MapK11. In chromatin immunoprecipitation assays, we used a cocktail of anti-FoxP2 antibodies to pull down FoxP2 bound to fragmented chromatin. Upon isolating the fragments, we performed RT-PCR using primers for MapK11. As predicted by our study, the fragment corresponding to MapK11 that encompasses the transcription binding site was identified. This data has been added to Figure 6 as Panel D and is now included in subsection “A bioinformatics approach indicates MAPK11 as an entry point to neuromolecular networks for vocal learning”.

Additionally, the drawing of connections between MAPK11, ATF2, and H2B needs further support to be persuasive. This is especially apparent in the reasoning concerning H2B. Histone H2B is a core nucleosome component, constitutively expressed across tissues. Its post-translational modification has important roles in regulating transcription, but it's unclear if regulation of its expression per se has roles in modifying the expression of specific genes. The argument here is that H2B expression levels are modulated in response to singing due to the gene's inclusion in the blue network. The authors include a citation in support of this argument. However, the paper cited describes H2B acetylation differences in a learning paradigm and not expression differences, which would be the more direct comparison.

Remarkably, in experiments conducted in *Aplysia* more than ten years ago, Kandel, Schwartz and colleagues proposed that the pathway from MAPK11 to histone acetylation via phosphorylated ATF2 mediates short and long-term synaptic depression (LTD) in addition to epigenetic changes. Moreover, mice expressing the Foxp2 mutation that disrupts language in affected KE family members perform poorly on a wheel running task compared with controls. This task relies on striatal LTD. Accordingly, LTD is deficient in these mice, and the dendrites on medium spiny neurons are reduced (Groszer et al., 2008). Conversely, mice that express the normal human FoxP2 isoform exhibit enhanced sensorimotor learning, enhanced LTD, and increased dendritic spines on MSNs (Enard et al., 2009; Schreiweis et al., 2014). LTD within Area X is one of the few forms of synaptic plasticity demonstrated in song control circuitry (Ding et al., 2003). Together with our network data, these observations link FoxP2 to altered Area X function via a broadly conserved pathway for learning and memory. We now cite that publication (Guan et al., 2003) in addition to citing the rodent learning and memory work regarding an interaction between MAPK11, ATF2 and H2B.

Still, we appreciate the point that the reviewer is making: we cite prior evidence that indicates the protein-level interaction between ATF2 and H2B is the driver for a learning behavior. Our own data indicate that H2B expression is positively correlated to singing but we do not present evidence supporting its acetylation being related to learning in our animals. This speaks to a point we make in the Discussion section and in main text Figure 7 and 8. Gene co-expression does not indicate functional interaction between proteins. Our statements regarding the acetylation of histone H2B are a hypothesis about an interaction that is underlying vocal learning that is based upon the combination of prior work in other model systems and what our network data suggest to be relevant. The interaction between ATF2 and H2B serves as the functional link between “do singing” and “do learning” co-expression that co-occur in 65d and not adult Area X. We use this principle to guide the construction of the protein interaction network presented in Figure 7 and generate hypotheses en-masse. There are multiple other pathways in our protein interaction network analogous to ATF2 and HISTH2B, where a well-connected member of a learning module interacts with a singing related gene. The advantage of using networks to guide the interpretation of transcriptomic data is clear here: The prioritization of nodes by their network position allows for molecules with no a priorirelationship to a trait to ‘rise to the top’ and reveal themselves as worthwhile targets for investigation. Our data-sharing plan is to make these interactions publicly available upon publication of our manuscript.

Reviewer #2:This study examines the relationships between striatal expression of FOXP2 isoforms and vocal learning and variability in a songbird. This is an important contribution, in particular because this gene's involvement in vocal learning cannot be properly studied in models like mice, which lack this behavior. The study takes advantage of the knowledge on vocal learning and circuitry in finches and makes extensive use of weighted co-expression analysis across ages and basal ganglia subregions. It contributes novel data and insights, indicating that different gene sets are co-expressed in correlation with different states of singing or learning. It also points to novel relationships among genes, and candidates for mechanistic studies. The effort is a tour de force, the expression data are impressive, and the findings seem robust.

We appreciate the thoughtful comments of reviewer 2, including that our effort is a 'tour de force, the expression data are impressive, and the findings seem robust'.

One issue is that the study does not provide direct evidence that the major variant examined, the short 10+ isoform, occurs endogenously in finch brain tissue. The fact that this variant transcript exists is not sufficient, as transcripts do not reliably reflect expression at protein level. Because this point is novel and central, the authors should consider options for demonstrating the endogenous occurrence of this protein isoform, such as mass spectrometry to detect the isoform's unique peptide or developing a specific antibody. The lack of direct evidence weakens a major underlying assumption of the study and will require toning down several statements about expression of that isoform and the implications for many of the datasets presented.

In response to the reviewer’s query, we now provide evidence in the form of an immunoblot that indicates endogenous expression of the truncated FoxP2 isoform in zebra finch. Briefly, 5 or 10 ug of adult whole brain homogenate were probed with an antibody against the N-terminus of FoxP2 (ProteinTech Cat. No. 20529-1-AP). Bands are revealed at ~72kDa, reflecting the fulllength isoform, and at ~55kDa reflecting the truncated form. This is now included as Figure 1—figure supplement 1 and mentioned in subsection “Virus-mediated overexpression of FoxP2 isoforms affected song learning and/or vocal variability”.

With regards to MAPK11, the authors' inference about the putative transcription start site (TSS) and promoter definition is likely incorrect. In silico predictions are often incomplete and need to be taken with caution in the absence of data like cDNAs with intact 5' caps, or primer extension assays. The orthologous MAPK11 RefSeq models from other species (chicken, mammals, Xenopus) do not map to the zebra finch genome for their first 100-200 nucleotides (readily verified with alignments in UCSC's genome browser). In fact, this 5' segment in other species corresponds to a first exon that maps several kb upstream of the second exon for this gene in the respective genomes (~12 kb in chicken, where the gene has 12 exons). Assuming the study was based on Ensembl's prediction in zebra finch (please indicate if otherwise), this first exon is not predicted: the model has only 11 exons and does not have a starting ATG, indicating 5' end incompleteness. A 5'-most exon, if found, would bring the exon number to 12 and likely lie several kb upstream (possibly in a gap) to the first exon of the Ensembl model. Thus, the authors have most likely not identified the correct TSS/promoter region for this gene in finch. Further, to assess the significance of the occurrence of any given DNA binding motif, an estimate of chance occurrences in the genome needs to be provided. In sum, the claims about the presence of a possible FOXP2 binding site(s) in promoter region of MAPK11 are not supported by the evidence provided.

In the subsections “A bioinformatics approach indicates MAPK11 as an entry point to neuromolecular networks for vocal learning” and “Transcription Factor Binding Site Analysis”, we define the promoter region as the first 1000 bases upstream of the transcription start site, as defined by RefSeq. Our study is based on RefSeq predictions, which places MAPK11 on Chromosome 1A (NC_011463.1) with 11 exons. RefSeq also predicts the translation start ATG within the first exon at base 19646171. We agree that in silico validations need to be interpreted with caution, so in response to this concern, and to a related concern of reviewer 1, above, we conducted a direct functional test of the predicted interaction between FoxP2 and the MAPK11 promoter. In chromatin immunoprecipitation assays, we used a cocktail of anti-FoxP2 antibodies to pull down FoxP2 bound to fragmented chromatin. Upon isolating the fragments, we performed RT-PCR using primers for a region of the MAPK11 promoter. As predicted by our study, the fragment corresponding to MAPK11 that encompasses the transcription binding site was identified. This data has been added to Figure 6 and is now included in the subsection “A bioinformatics approach indicates MAPK11 as an entry point to neuromolecular networks for vocal learning”.

[Editors' note: the author responses to the re-review follow.]

[…] Below is a list of the most important points, consensus from extensive discussions among the reviewers after they have all submitted their individual reviews.1) As pointed out by reviewer 2 (the new reviewer), echoed by previous rounds of both reviewers 1 and 3 and remaining concern of reviewer 3, a major issue is whether your data provide unequivocal support to the notion that FoxP2 manipulations affect area X gene expression and gene network (a major claim of the manuscript that is reflected in the title). This 'learning network' is present in both control (GFP) and all animals combined and is generally correlated with tutor song similarity across groups, but there is no evidence that it or any other network was affected by the FoxP2 manipulations. Short of asking you to redo the experiment of analyzing gene expression in only virally transduced regions, you are strongly encouraged to dig further into the analyses comparing specific genes, including more qRT-PCR on control genes to find that maybe there is a viral FoxP2 manipulated gene expression signal in the Area X punches. If you cannot find gene expression changes due to FoxP2 manipulations, you should state clearly the results, substantially modify this part of the manuscript, including the title.2) All reviewers expressed frustrations that much of the data and analyses you described in the rebuttal did not make into the revised manuscript. Please include useful materials in revised manuscript, at least in supplemental figures, so that readers can have additional perspectives.3) You need to spell out more clearly in the revision (not just the rebuttal) where you did/did not correct for multiple comparisons and rationalization for including 'non-significant' correlations in further analysis.4) Reviewer 3 has substantial concern whether the data allow you to claim that FoxP2 binds to in the MAPK11 promoter rather than intron. Calling it a regulatory region might be more appropriate.

Overview of major revisions:

1) We built and compared construct-specific networks from birds injected with either the FoxP2.FL-expressing virus (FoxP2.FL) versus those injected with the FoxP2.10+ expressing virus (FoxP2.10+) versus those overexpressing GFP (GFP). We quantified module preservation between the FoxP2 networks and the GFP network. In both FoxP2 networks, a gradient of module preservation was observed versus the GFP network with both overlapping and significantly different modules observed. These results directly address the question as to whether the viral constructs are differentially affecting gene expression: They are. Had the individual FoxP2 networks been strongly preserved with the GFP network, no effect of virus could be assessed. These findings are now provided as Figure 3—figure supplement 2 to Figure 3—figure supplement 5.

2) We have substantially increased the number of figure supplements in this revision to include many of the figures previously found only in the response to reviewer’s letter.

3) We have added a new paragraph to the “Correlation of behavior to gene expression” section to the methods wherein we detail corrections for multiple comparisons and the rationale behind including non-significant p-values after FDR correlations in downstream analyses.

4) We have updated the manuscript to remove any references to a MAPK11 promoter and synthesized reviewer 1’s helpful comments into a new Materials and methods “MAPK11 Annotation Note” subsection.

Reviewer #1:The authors have revised the manuscript substantially, addressing several of the main issues that were previously raised. This includes a re-analysis of the 'learning-associated' module in control birds, as well as further data showing the endogenous expression of the truncated FOXP2 protein isoform and providing evidence for binding of FOXP2 binding to a predicted site in the MAPK11 candidate target gene. The authors also provided clarifications on several other important methodological and interpretation points. The manuscript is thus much stronger and better supported by the data.

We thank reviewer 1 for his/her positive comments about the revision.

Some concerns remain, however. The authors have not directly addressed the multiple comparisons and statistical significance. They acknowledge the issue in the rebuttal, but apparently have not made significant related adjustments in the paper, and still show all the correlations performed. Arguably the study lacked statistical power to perform all these multiple comparisons and correlations, given the sample size and the scope of the effort. The authors state, though, that some correlations were stronger and remained significant even after FDR corrections. Thus, perhaps a reasonable compromise would be to indicate which correlations survived the FDR correction, and suggest that others might be significant with a larger n and/or different design.

In response to this concern and as suggested by reviewer 3, we have denoted in the figure legend for Figure 3 that we present uncorrected p-values, then direct the reader to subsection “Correlation of Gene Expression to Behavior” wherein we provide rationale for our statistical approach, summarizing what was said in response to the first comment made by reviewer 2 (now reviewer 1) in the previous review.

[…] Rather than having identified the TSS and thus the promoter region for zebra finch MAPK11, it is much more likely that the authors' data provide evidence for FOXP2 binding at an internal, presumably intronic site. This is quite interesting in itself, and consistent with data from other genes and organisms that relevant transcription factor binding sites are not restricted to promoter regions. The author's interpretation that they have identified the promoter region for MAPK11 is less tenable without further data, as it would imply that this gene in zebra finches has a considerable truncation at its 5'-region, resulting in the loss of a N-terminus peptide that is conserved across vertebrates, including other closely related songbirds and mammals. These issues should be acknowledged and the claim that the TSS/promoter has been identified needs to be toned down.

We thank reviewer 1 for this thoughtful analysis and presentation of the data and agree that we have not provided sufficient evidence to suggest that we are looking at the zebra finch MAPK11 promoter region. In this revision, we have removed such claims, acknowledged that the RefSeq model for zebra finch MAPK11 is likely incomplete, and suggested that we are perhaps viewing a regulatory region within MAPK11 that is bound by FoxP2 instead of the promoter. We now refer to this region as the “promoter”, quotes included. Finally, we have added a new subsection ‘MAPK11 Annotation Note’ summarizing these helpful comments in the Materials and methods section and referenced it in the text.

It would also be helpful if the authors could deposit the sequence(s) identified in the ChIP assay for FOXP2 in GenBank, to allow for precise mapping onto the zebra finch genome.

We sequenced a genomic fragment that was pulled down by FoxP2 ChIP and found it aligned well with the NCBI sequence for MAPK11. We provide that sequence here as Figure 6—figure supplement 1. The sequence is less than 200bp and thus GenBank will not accept it.

Reviewer #2:I was asked to review the original version of this paper but could not at the time due to time constraints. However, I reviewed the revised version, and did so first without looking at the original version or the reviewer's comments on that version. So this represents a fresh view of the revised paper. I only looked over the prior reviews and reviewer comments afterwards.I think the underlying question, experiments, and raw data generated in this manuscript is very good. This includes finding that 2 different isoforms of FoxP2 have different functions in the striatal song nucleus AreaX for song production and song learning, and that there are differences as well as overlaps in the song production and song learning gene regulatory networks of juveniles versus adults.

We appreciate Dr. Jarvis’s fresh perspective on the paper and positive comments.

However, I see issues with the preparation of the paper, interpretations, and one of the important experiments on viral manipulation of gene expression as measured by RNASeq in Area X. For preparation, the manuscript is not presented clearly enough, causing confusion to the reader. Sometimes discussion is in the results and vice versa; figures are not explained well enough. For interpretations, an example is the words 'learning' and 'song', which are used too loosely, as opposed to 'song learning' and 'song production'.

We appreciate this point and have tried to clarify our use of the words ‘learning’ and ‘song’ throughout the manuscript. For example, we now refer to the ‘song-related’ modules as ‘song production’ modules since their co-expression patterns correlate with the amount of singing. When viewed strictly in Area X, the modules referred to as “learning” modules could be described as “song learning” modules, as the reviewer suggests, since their co-expression patterns correlate with song learning; a key finding of the paper. However, since these co-expression patterns exist outside of Area X, referring to them in that context as “song learning” modules may not be entirely accurate. We thus now refer to them as ‘learning-related’ modules throughout the manuscript. We hypothesize that in VSP, these ‘learning-related’ co-expression patterns underlie non-vocal motor learning and/or general plasticity. In line with this idea, we videotaped animals on the morning of sacrifice and used an observer to score the non-vocal behaviors. We did not find any correlation between these behaviors and gene expression patterns in the outlying VSP. We hypothesize that this is because none of those non-vocal behaviors were being actively learned.

Another is the function of the VSP, with is mostly a motor area, but not explained.

We apologize that we did not define VSP early in the paper. This is now remedied in subsection “Virus-mediated overexpression of FoxP2 isoforms affects song learning and/or vocal variability”. We have also added reference to Reiner et al., 2004 in subsection “Virus-mediated overexpression of FoxP2 isoforms affects song learning and/or vocal variability”. For context about the function of VSP, we rely on the work of Pfenning et al., (2014) and, more specifically, Feenders et al., (2008) which provide foundational evidence for the function of these regions, including VSP, directly adjacent to song control nuclei as being the neural substrates for non-vocal movement-associated pathways. These publications are cited in the reference section of our manuscript.

There are more of these examples highlighted in my specific comments below. The most worrying item is the RNASeq FoxP2 manipulation analyses. The authors skipped a step in the analyses in the paper, which is determining the differences of FoxP2 manipulation in the basal- and singing-driven gene expression of Area X. It looks like this might not be possible to do with their current data, because they punched biopsied of the entire Area X, and not the 25% of it that had the gene manipulation. They might have been able to see that 25% if they used a GFP marker under a fluorescent scope in their dissections of the tissue.

We agree that specific dissection of virus-infected cells marked by a GFP reporter could alleviate signal-to-noise problems. To achieve long-lasting overexpression of FoxP2, however, we used an AAV1 vector with a limited cloning capacity (~5 kb). In experiments not presented in this manuscript, viruses that expressed both FoxP2 and GFP separated by a P2A sequence, which reached 99.5% of the virus’s cloning capacity, yielded poor expression of the gene product and would have been useless for the behavioral experiments. In order to drive overexpression of FoxP2 to behaviorally relevant levels, our AAV1 vector had no tag on the FoxP2 transcript itself, making infected cells visually indistinguishable from uninfected cells.

Given that a single amino acid alteration in human FOXP2 leads to a severe speech and language disorder, we opted not to otherwise tag the zebra finch FoxP2 sequence that was inserted into the virus, avoiding any concern about altered function due to altered sequence. Another benefit of the construct was that it is the same as that used by Heston et al., (2015) enabling comparison and validation across studies.

What they did instead was group all the samples together (control and manipulated animals) and created a network expression analyses in Figure 3, with the assumption that it is affected by the viral vector manipulations in Area X. One would hope that there is a partial affect from the 25% of cells of the manipulated Area X on the RNASeq data, and this can be tested. It took me some effort in re-reading to figure out what the authors did.After reading the reviews, I see original reviewers 1and 2 had the same concern, just maybe not expressed as clearly. In response to that concern, the authors did a GFP manipulated Area X only analyses and came up with the same results (Author response image 1 of the response letter). This is good and should be included in the main paper.

As suggested, we have added the GFP-only network analysis to the main paper as (Figure 3—figure supplement 1). Module preservation statistics indicate co-expression patterns that are distinct from the other construct-specific networks. This finding supports our claim of construct-specific gene co-expression.

However, they did not show network analyses of the viral FoxP2 manipulated animals, but instead show that clustering of expression does not differ between the GFP and FoxP2 manipulated Area X (Author response image 7 of response letter). So, if we take their analyses at face value, this would mean that the combined network analyses of control and FoxP2 manipulated animals in Figure 3 of the manuscript may not be influenced by the FoxP2 manipulation, and instead is simply a control network similar to unmanipulated animals. This needs to be shown.

In addition to the GFP network, we now include individual network analyses for the FoxP2.FL and FoxP2.10+ groups. These are now included in the main paper as (Figure 1—figure supplement 2 to Figure 1—figure supplement 5). The key findings are:

1) Differential network analyses across the two FoxP2 (FoxP2.FL and FoxP2.10+) and GFP networks reveal a spectrum of module preservation with some modules being highly preserved across the three groups and others being poorly preserved. Notably, the green learning-related module is well-preserved across all groups. Birds in these experimental conditions were siblings, and in some cases from the same clutch, suggesting that the driving effect of network differences is the construct-specific manipulation.

2) The green learning-related module is preserved across the GFP and both FoxP2 networks and shows a strong correlation to learning (that passes FDR correction) in the GFP network.

These new observations have been added to the Results section of the revised manuscript.

Further, if it is not affected by the FoxP2 manipulations, then it is a flaw of the experimental design of the punch biopsy approach and the sentence, in subsection the “Gene modules in juvenile Area X that correlate to vocal behavior were enriched for communication and intellectual disability risk genes”: “We used RNA-seq to quantify gene transcription in Area X of 65d juveniles overexpressing GFP, FoxP2.FL or FoxP2.10+, then used WGCNA to identify gene co-expression modules and link them to song learning”). If a lack of a difference in control and FoxP2 manipulated animals is true, then I think the RNASeq experiments should be presented in another paper, because the potential FoxP2 manipulation would have been drowned out and this is still an interesting finding in the normal singing juvenile vs adult states. However, if a small affect is found in the noise of the 75% unmanipulated Area X, then that affect would belong in this paper.

In addition to the construct-specific network analyses noted above and presented in the present revision of this manuscript that indicate FoxP2 isoforms are altering gene co-expression patterns in Area X, we have shown in the main text that our FoxP2 viruses:

1) Differentially affect behavior, specifically vocal learning (FoxP2.FL) or variability-induction (FoxP2.10+) relative to birds injected with the GFP-expressing construct.

2) Drive overexpression of their specific isoforms, as assessed by qRT-PCR (Figure 1; recently replicated in a new time-course study; see Author response image 8 below).

3) Drive overexpression of their specific isoforms, as assessed by *in situ* hybridization (Figure 1).

Moreover, we did recover overexpression of GFP in the group of birds that received the GFP construct in our RNA-seq data. The lack of significant increases in FoxP2 expression in birds that received FoxP2 constructs (which only differ from the GFP construct in the inserted sequence: GFP versus FoxP2.FL versus FoxP2.10+) in the RNA seq data may be due to: (1) Weaker signal-to-noise (as construct-driven overexpression lies on top of endogenous, changing FoxP2 levels; which is not the case for GFP); (2) Insufficient sample size; (3) Poor sensitivity of RNA-seq vs. qPCR, and/or 4) in vivo-down regulation of endogenous FoxP2 to compensate for exogenously-driven levels.

To check for the latter possibility, we conducted a new time course study. As in the original study, young birds were injected with the FoxP2.FL overexpressing construct at ~30 days of age. They were then sacrificed (in the morning at lights on, before any singing) at 10-day intervals up to ~60 days of age. Area X was again micro-punched and FoxP2 levels were measured by qRT-PCR. Compared with uninjected age-matched controls, we found significant elevation of FoxP2 levels in Area X (blue bars in the below graph) at 10, 20 and 30 days post-injection including at the ~60 day old time point. This argues against endogenous down-regulation (#4 above). As in the original manuscript, no significant elevation of FoxP2 levels was observed in outlying VSP (yellow bars in the below graph). In sum, this confirmatory result conducted on 3 birds per time point/condition replicates our prior qRT-PCR analysis and shows significantly elevated FoxP2 transcript expression in birds that receive the FoxP2- construct relative to uninjected controls.

**Author response image 8. respfig8:** FoxP2 expression in Area X is elevated for up to 30 days following AAV-FoxP2 injection. Juvenile birds (~30d) were injected with our FoxP2 overexpressing construct, bilaterally into Area X. FoxP2 levels were examined 10 days (n=3), 20 days (n=4), and 30 days (n=3) post-injection. Results were normalized to uninjected birds sacrificed at ~60 days of age (n=3). Area X FoxP2 levels (blue bars), were significantly elevated at 20 days post-injection and remained so at 30 days post-injection. No such increases were observed in outlying VSP (yellow bars).

Coupled with (a) the construct-specific *in situ* hybridization data, (b) the documented GFP overexpression and, most recently, (c) the differential module preservation across the 3 construct-specific networks and, most importantly, (d) the behavioral differences among the 3 construct-specific groups, we conclude that the RNA-seq analysis that we performed was not sensitive enough to overcome signal to noise issues (e.g. low n per group, measurement on top of endogenous levels). In contrast, the qRT-PCR analysis was sensitive enough to detect overexpression in both the original samples and now in new samples, despite a low sample size and ~25% transfection rate.

In preparation of this manuscript, we considered the possibility of presenting these data as two separate manuscripts, one in which we discuss the behavior and the other in which we discuss the gene expression data. Rather than do so, we reasoned that a great strength of our approach lies in linking gene expression with behavior. Indeed, behavior guides the focus on and interpretation of the network modules. For example, based on network analysis alone (not considering behavior) the green learning-related module is densely interconnected and of great network importance in this study. We strongly feel, however, that what is really exciting is its correlation with learning in juveniles. This gene-behavior link provides key context for understanding its poor preservation in adulthood.

Reviewer #3:The authors have been very responsive in their revisions and have made several key additions and clarifications that substantially improve the manuscript. Specific comments are included below regarding these main points. The residual issues raised below are ones that the authors should be able to address in Discussion section or with modest additional work.

We appreciate reviewer 3’s enthusiasm for the updated manuscript.

1) Particularly nice is that they have found a correlation between the "learning network" and tutor song similarity for the GFP control group of birds. This significantly addresses one of my primary concerns in the initial submission regarding whether the network that was identified in FOXP2 manipulated birds would be present and correlated with learning in a control population. It is impressive that the green module was recovered in the GFP-only samples and that its correlation with tutor similarity was maintained. I find this result particularly interesting in conjunction with the analysis in Figure 6, showing that MAPK11 levels perfectly correlate with tutor similarity within virus conditions.2) An important related addition is the PCR analysis of FOXP2 expression levels in ventral striatum. These data further support the authors contention that the presence of the learning network in both Area X and ventral striatum is not a direct result of the viral manipulations (spillover to ventral striatum) in both nuclei.3) The further elaboration regarding MAPK11-ATF2-H2B potential link to learning is also a good addition that is more clearly grounded in previous work and seems less speculative.4) Related to 3: the intent behind the FOXP2 ChIP experiments is excellent, and such data could potentially provide further support the authors' conclusions. However, to be compelling they would require two additions:a) Presentation of qPCR instead of (non-quantitative) endpoint PCR data.b) Inclusion of at least 2 negative control regions to enhance the evidence for specificity (The IgG control is good but does not control for within-antibody noisiness).

We performed the ChIP-qPCR as requested including two negative controls, β actin and 3000 bp upstream of Cntnap2. β actin was chosen based on previous FoxP2 ChIP-chip experiments in embryonic mice (Vernes et al., 2011). FoxP2 has been shown to bind to the promoter region of Cntnap2 (Adam et al., 2017), thus moving further upstream from the promoter region was determined as a negative control to demonstrate proper sonication fragment length as well as a region close to a known binding site. These negative controls show the specificity of our antibody pull down. Additionally, qPCR data highlighting our proposed FoxP2 binding motif is shown by the MAPK11 primers. We chose to show the primers covering the region of the binding motif and the adjacent regions. These results confirm FoxP2 binds to this region of the MAPK11 gene.

**Author response image 9. respfig9:** FoxP2 ChIP-qPCR shows an enrichment around the putative MapK11-FoxP2 binding motif. To confirm our ChIP-PCR result, negative controls β actin (-500 bp upstream of TSS) and Cntnap2 (-2500 bp upstream of TSS of Cntnap2) show no enrichment of FoxP2 pulled down DNA by qPCR. Alternatively, primers amplifying the proposed FoxP2 binding motif and the adjacent regions, show increased signal, confirming our ChIP-PCR result.

I think the authors ought to be able to provide these additional data without too much additional work. If, however, this is problematic, an alternative would be to exclude these data and instead expand the bioinformatic analysis as suggested in the previous round of reviews to assess the statistical significance of the FOXP2 motif upstream of the MAPK11 TSS (I.e. to judge how likely this is by chance.)5) The clustering analysis of the samples by their expression profiles is good and should be included in the text.

We have added this as Figure 3—figure supplement 6.

6) General Interpretation. The revised manuscript more firmly establishes an association between the 'green network' and tutor song similarity. This association rationalizes the use of 'learning network' to refer to the green network. However, the authors should be clear in their discussion of this association that these data are primarily correlative and the role of this network in learning per se remains speculative and a subject for future investigation.

We have updated the first paragraph of the Discussion section to further denote that the associations observed in this manuscript are correlative.

I think that the authors intend to make the point that the presence of this network in the VSP indicates that it may capture more general (not song specific) aspects of nervous system development or plasticity that correlate with song learning; if so, this could be made more explicit in Discussion section.

We have updated the third paragraph of the Discussion section to underscore this important point. We also make this point graphically in Figure 8.

Along those lines, it seems that such generalized changes could arise (and be correlated with tutor song similarity) in a variety of ways, which could reflect more or less direct involvement of this pathway in song learning. For example, both song learning and general ns development (captured by green network) might depend on a third variable (hormonal status, variation in neuromodulatory tone, social interactions and stimulation associated with effective tutor song experience, etc.). Indeed, it even seems possible that tutor song similarity achieved by different individuals itself feeds back to influence social and hormonal status of birds in ways that promote or inhibit more general aspects of nervous system development, or plasticity networks that could be captured by the green network. The authors need not belabor these points, but I think they would be well served to at least briefly discuss alternatives to the possibility that variation in the green network is tightly linked to, and is causally driving, variation in song learning (which is how the treatment currently reads to me). In this context, the mechanistic investigations of network identity/membership/function (e.g. MAPK11-ATF2-H2B) are nice in that they illustrate how candidate mechanisms can be identified through network analysis and begin to build a stronger case for specific mechanisms.

Our intent was not to indicate that the green learning-related module is the driver of the vocal learning process. Instead, we present the connectivity of genes within the green module as an indication of being in a state of striatal learning, be it vocal (in Area X) or non-vocal (in VSP). The biological factors that lead to the co-expression/connectivity within the green module go beyond the scope of this manuscript but presumably those factors are what cause the formation of this module in juveniles and its dissolution in adults. As reviewer 3 points out, the MAPK11-ATF2-H2B pathway is an example of how the network can be mined for mechanisms underlying the behavior. Further analysis of these genes and how their role in the network changes with age will allow for piecing together the factors that control the network topology and, we believe, the behavior.

7) In general the authors addressed many issues in reasonable fashion in the rebuttal but for many of them did not address these points in the revision. In some cases the authors note that the requested analyses were not included in the revision because 'the results were negative', but in these cases a negative result is informative and effectively addresses potential concerns about interpretation of the primary data. Where these data (in rebuttal) address concerns raised by the reviewers they are also likely to be of interest to readers of the published manuscript, and as much as possible the authors would be well served to include most of these analyses in the revised manuscript – perhaps in supplementary material.

As noted above, we have included many of the figures previously found only in the first response to reviewer letter as supplementary figures.